



# Eddy-covariance with slow-response greenhouse gas analyser on tall towers: bridging atmospheric and ecosystem greenhouse gases networks

Pedro Henrique H. Coimbra[1,2*], Benjamin Loubet[1], Olivier Laurent[2], Laura Bignotti[1], Mathis Lozano[2,] Michel Ramonet[2]

[1] ECOSYS, INRAE, AgroParisTech, Université Paris-Saclay, Palaiseau, France
[2] Laboratoire des Sciences du Climat et de l'Environnement, CEA, CNRS, Université Paris-Saclay, Gif-sur-Yvette, France

* corresponding author: pedro-henrique.herig-coimbra@inrae.fr

**Abstract**

Greenhouse gases monitoring is important to ensure climate goals are being achieved. This study unveils the potential of using atmospheric tall towers in direct flux measurements, bridging the gap between atmospheric and ecosystem monitoring networks. The ICOS Cities (PAUL) project aims to monitor $CO_2$ emissions in urban areas, where concentrated emissions make them key targets for climate change mitigation. This study explores synergy between ICOS atmospheric and ecosystem networks by utilizing slow-response analysers (~2 sec) on tall atmospheric towers for ecosystem studies using the Eddy Covariance method. A standard setup with an ultrasonic anemometer and an infrared (IR) fast-response $CO_2$ analyser was installed and compared with measurements from an existing cavity ring down spectroscopy (CRDS) analyser measuring $CO_2$, CO, and $CH_4$. Deployed on the 100 m Saclay tower near Paris, covering a 43.9 km² 80% footprint with heavy traffic roads, a nearby heating plant, and a forest, the setup addressed technical challenges and height-induced complexities. Corrections for flux attenuation by high frequency losses were limited to <20% on average for all stabilities, around 11% for unstable conditions. Wavelet-based eddy covariance allowed 18-34% more data exploitation than standard EC enabling the analysis of non-stationary fluxes, particularly from a point source such was the case of a heating plant. The estimated storage term produced by atmospheric profiling measurements reported an expected increase at night, destocking during the first half of the day. Storage term represented at times more than half of the surface flux. Elevated mean fluxes for $CO_2$ (10 $\mu mol\,m^{-2}s^{-1}$) and $CH_4$ (200 $nmol\,m^{-2}s^{-1}$) were observed from the heating plant wind direction during December and January. Conversely, the forest direction exhibited the strongest sink among all wind directions, with −4 $\mu mol\,m^{-2}s^{-1}$ during July and August. These results demonstrate the feasibility and versatility of utilizing atmospheric towers for urban emission monitoring, offering valuable insights for emission monitoring strategies worldwide.



## 1. Introduction

Global surface temperature is 1.6 °C warmer on land compared with the pre-industrial era (IPCC, 2021), and projections show

more than 2°C warming in 2100 (IPCC, 2022). Warming results from the increase in greenhouse gas (GHG) concentration in the atmosphere, mainly driven by anthropogenic emissions (IPCC, 2021), of which 86% comes from fossil fuel $CO_2$ (Canadell et al., 2021). Agriculture, forestry and other land use (AFOLU) are a significant source of GHG (12.0 ± 2.9 GtCO2eq yr−1), while concurrently possessing the potential to remove $CO_2$ from the atmosphere (Jia et al., 2019).

Urban areas concentrate human activities and represent a significant source of GHG emissions, consequently making it one of

the targets for mitigating climate change. Many northern countries' cities have ambitious GHG emission reduction plans over the next 2 decades, that consist of electrifying the energy grid, implementing car-free zones, and investing in insulation improvement. Consequently, there arises an imperative for robust monitoring of urban areas' emissions reduction. Several works have tried to decompose eddy covariance measurements in (sub-)urban setup with different degrees of uncertainty (Velasco et al., 2009; Bergeron and Strachan, 2011; Ueyama and Takano, 2022). Currently in Europe the project ICOS Cities

(PAUL) aims to advance technologies for monitoring $CO_2$ concentrations in urban areas of three different sized pilot cities (Munich, Paris, and Zurich).

Monitoring GHG in the atmosphere, ocean and ecosystem is the objective of world-distributed research infrastructures such as ICOS in Europe (Heiskanen et al., 2022). To that purpose, different methods are used on terrestrial sites. Ecosystems sites focus on local flux monitoring using high-frequency measurements, while atmospheric towers measure precisely the

concentrations as an imprint of larger scale fluxes. Ecosystem sites measure surface fluxes that represent a specific biome, as determined by the tower's footprint. In contrast, atmospheric sites have a footprint spanning several hundreds of $km^2$ and may be used to identify anomalies in $CO_2$ surface fluxes based on concentration (Ramonet et al., 2020) or retrieve surface flux by inverse modelling eventually using tracers (Ciais et al., 2011).

At local scale, Eddy Covariance (EC) is the reference method for GHG monitoring. The method is praised for directly and

continuously measuring surface turbulent flux and largely applied since early measurements to different gases, including water vapour, $CO_2$, $CH_4$ and $N_2O$ (Valentini et al., 1996; Moncrieff et al., 1996; Fowler et al., 1995). Standard measurements require fast-response instruments, which is a technical limitation for measuring certain compound's concentrations. Long-term measurement sites are equipped with $CO_2$ and $H_2O$ gas analyser, and in some wet or agricultural sites with $N_2O$ or $CH_4$ analysers (Nemitz et al. 2018). At larger scales, atmospheric concentration measurements are often used alongside meso-to-

continental scale transport models to solve surface flux (Lauvaux et al., 2012). This top-down approach is often validated locally by direct EC measurements (Vuichard et al., 2016).

The differences between typical atmospheric and a flux tower monitoring setup is that: (1) atmospheric towers are taller (above 100-meters height) whereas flux towers range from 2 to 40 meters height. This is because atmospheric measurements are setup to catch the seasonal and annual trend in atmospheric background concentrations at regional scale which requires limiting the

impact of local sources (Yazidi et al., 2018); (2) atmospheric towers have more precise measurements but slower, not cadenced,





sampling rate around a few seconds, whereas EC sample from 5 to 20 Hz. This is required by the eddy covariance method; (3) ecosystem stations have ecosystem monitoring (vegetation and soil sampling), which are not measured in atmospheric stations, while these measure additional gaseous compounds ($CH_4$, $N_2O$, CO) (Hazan et al., 2016).

Being able to use slow-response analysers to calculate flux by eddy covariance has been identified as a useful strategy to
expand the flux networks to other compounds (Wohlfahrt et al., 2009). Atmospheric towers have high precision analysers which, if we can use them to compute Eddy-Covariance fluxes, would provide multi-species flux measurements that would expand the flux network. This would require a fast 3D anemometer and continuous data logging at these sites. However, the constraints for concentration and flux measurements are not the same, and so not all towers may be suitable. For any atmospheric tower a couple of adversities must be addressed first. Discarding atmospheric stations on mountains that have
unsuitable conditions for flux measurements, we focus on tall tower over reasonably flat landscapes:

- Firstly, measuring flux with a 3s-response time analyser is challenging. Indeed, fast-response analysers, typically with 100 millisecond response time, are needed for flux measurements to capture the small and fast eddies (turbulent fluctuations) that carry most of the flux signal in the surface layer (Kaimal and Finnigan, 1994). Using slow-response analysers on short towers would mean losing most of the signal. For instance, the frequency with the highest
contribution to the flux on a 4- and 37-meters tall tower was 0.16 Hz (6 s period) and 0.02 Hz (50 s period), respectively (Coimbra et al., 2023). Using a slow analyser in these towers would attenuate the flux by 65-80 % and 30-45 % respectively in unstable conditions with wind speed from 3 to 7 m/s, and even greater attenuation is expected in stable conditions. Fortunately, the contribution of higher frequencies to the EC flux is inversely proportional to height (Horst, 1997), and so for the same unstable conditions measurements at 100 m would give a peak contribution
between 0.002 and 0.009 Hz (8.3 and 1.85 minutes period) and the high-frequency attenuation would therefore be small (10-20%). High frequency (HF) corrections based on predefined or experimental cospectra profiles are well established and routinely applied to correct for tube attenuations in ICOS and other flux networks (Horst, 1997; Massman and Lee, 2002; Ibrom et al., 2007; Fratini et al., 2012). We therefore expect sampling with slow-response analysers at tall tower may be suitable because the peak of the covariance cospetrum would be well caught and could
be corrected with standardised approaches (Massman, 2000).

- Secondly, the height also affects the source area. Taller towers have bigger footprint and often higher heterogeneity, commonly including both artificial and vegetated patches in the same wind sector. Heterogeneity and point sources can induce sudden shifts in the concentration due to wind direction changes, which will later be flagged as non-stationary by standard eddy covariance procedures. This quality filtering results in the loss of a significant amount of
data for less stationary surface fluxes, such as $CH_4$ and $N_2O$ (Irvin et al., 2021; Mishurov and Kiely, 2011). Whereas standard EC requires stationarity, wavelet-based EC does not (Torrence and Compo, 1998; Mallat, 1989; Farge, 1992; Farge and Schneider, 2001). Wavelet-based EC methods are sought in airborne campaigns when short-time resolution is needed (Strunin and Hiyama, 2004; Mauder et al., 2007; Desjardins et al., 2018; Metzger et al., 2013) and have been used to retrieve outbursts and non-stationary flux (Schaller et al., 2017; Göckede et al., 2019). By not requiring





stationarity, they yield high-quality data with fewer gaps. In two ICOS ecosystem sites Wavelet-based EC methods have been found to increase by 17 to 29 % the number of high-quality observations (Coimbra et al., 2023). The number may even be greater in urban conditions where point source and denser landscape would enhance surface fluxes heterogeneity.

- Lastly, the height also require accounting for storage fluxes below the EC measurement height, and vertical advection
fluxes components (Aubinet et al., 2005). Storage flux arises from the accumulation or release of the compound below the measurement height. The flux at the ground is hence the sum of the flux at the measurement height and of the storage flux below. Positive storage flux may result from the decoupling of surface and atmospheric dynamics. Such a decoupling may arise especially under stably stratified surface layer, occurring at night above canopies especially under radiative cooling conditions (Kaimal and Finnigan, 1994). Negative storage fluxes arise during the early
morning when the atmospheric boundary layer raises and the stably stratified layer breaks down (Aubinet et al., 2005). At tall towers the storage can be high and remain large in the morning when the vegetation starts photosynthesizing but the turbulence is still low (Haszpra et al., 2005). At very tall towers (300 m), the storage dominates the flux dynamics (up to 95% of the total flux, Winderlich et al. 2014). It should be noted that under ideal surface homogeneous conditions, the storage term is expected to tend to zero when averaged over a day and hence only affect
the surface flux dynamics but not the integrated fluxes.

In atmospheric towers, ICOS focuses on measuring not only $CO_2$ but also CO and $CH_4$ concentrations routinely. Therefore, measuring fluxes on these towers potentially enables the measurement of CO and $CH_4$ fluxes in the surrounding areas of each tower.

On mid latitudes in the North hemisphere most of CO emissions (14,000 $TgCOyr^{-1}$, 54%) come from direct (fossil or bio) fuel
combustion (Zheng et al., 2019). We expect this emission to increase during winter due to the diminished efficiency of fuel combustion induced by colder temperature (Helfter et al., 2016). Additionally, a substantial contribution to CO levels stems from the chemical oxidation of $CH_4$ (900 $TgCOyr^{-1}$, 40%) and volatile organic compounds (VOCs) (300 $TgCOyr^{-1}$, 12%) (Zheng et al., 2019). This oxidation process makes vegetation an indirect CO emitter through the release of biogenic VOCs, but this production is not local and would not appear as a flux from the surrounding of the tower. Soil do also emit VOCs, but
are up to three orders of magnitude lower than canopy emissions under usual conditions (Peñuelas et al., 2014). On the contrary, soils are mainly recognized as a CO sink (15 times stronger than soil source), primarily attributed to microbial oxidation processes (Inman et al., 1971; Conrad and Seiler, 1980; Conrad, 1996).

Globally for the 2008–2017 decade, the majority of $CH_4$ emissions arise predominantly from wetlands natural emissions (~24%), enteric fermentation and manure (~17%), and fossil fuels (~17%) (Saunois et al., 2020). Wetland emissions are
concentrated in tropical and southern regions (< 30∘ N), while fossil fuels are the predominant source in mid-latitudes (30–60° N). Agricultural waste contributes significantly in both tropical and mid-latitude areas (Saunois et al., 2020). In-situ observations showed considerable emissions from marshes (41±21 $gCm^{-2}yr^{-1}$), lakes (28±33 $gCm^{-2}yr^{-1}$), swamps (26±20 $gCm^{-2}yr^{-1}$), and fens (20±16 $gCm^{-2}yr^{-1}$) (Delwiche et al., 2021). The high emissions from marshes and high variability for



lakes highlight the dependence of sediment for $CH_4$ emissions. In a high flux lake (JP-SwL, 67 $gCm^{-2}yr^{-1}$), emissions can vary from a monthly average of 0.1 $\mu molm^{-2}s^{-1}$ to 0.3 $\mu molm^{-2}s^{-1}$ from winter to summer (Iwata et al., 2020). Fossil fuels observations from a 190 m tall communications tower in the centre of London showed a mean annual $CH_4$ flux of 46.5 $\pm$ 5.6 g C $m^{-2}$ $yr^{-1}$, increasing in the winter attributed to a seasonal increase in natural gas usage (Helfter et al., 2016). With that said, soils not only produce $CH_4$ but also consume it. Indeed, oxidation in soils is the primary inland process for $CH_4$ consumption (Canadell et al., 2021), making upland soils a net sink (Dutaur et Verchot 2007).

In this study, we evaluate the capability of using atmospheric monitoring tower with slow response analyser supplemented with a sonic anemometer to compute surface fluxes of $CO_2$, $CH_4$ and CO. To that purpose we installed a standard eddy covariance setup for $CO_2$ and $H_2O$ at 100 m at the ICOS FR-SAC atmospheric tower in the south of Paris, collecting 4 months of data starting from July 2023 until October 2023. The chosen site is a sub-urban site surrounded by a mix of agriculture, forest, wetlands, roads and buildings area. We then computed net $CO_2$ flux for slow and fast-response analysers and compared them. The high-frequency losses were determined and the correction procedure evaluated. The fluxes were calculated using the wavelet-based eddy covariance method detailed in Coimbra et al. (2023), while the storage flux was computed using three-point profile concentrations routinely measured at the ICOS tower. The seasonal variations and variations with wind directions of the $CO_2$, $CH_4$ and CO fluxes were then discussed.

## 2. Material and methods

### 2.1. Site description

The study uses data from a 100-meter tall tower in the French Alternative Energies and Atomic Energy Commission (CEA) at a research campus in Saclay, 20 km southwest Paris (Figure 1). The tower is part of the ICOS atmospheric network (FR-Sac) and takes part in the ICOS Cities, Pilot Applications in Urban Landscapes (PAUL) project, focused on integrated city observatories for greenhouse gases. Climatically, the area is under oceanic influence with mild temperatures (11.5°C annual mean) and moderate precipitations (677-700 mm annual). The surrounded landscape is dominated by artificial (buildings, roads), agriculture (mainly cereal) and forest. The region serves as a pathway for urban-to-suburban daily mobility with more than 60 thousand vehicles every day in 2022 according to SIREDO in the national (N118) and regional (D306, D36, D128) roads.


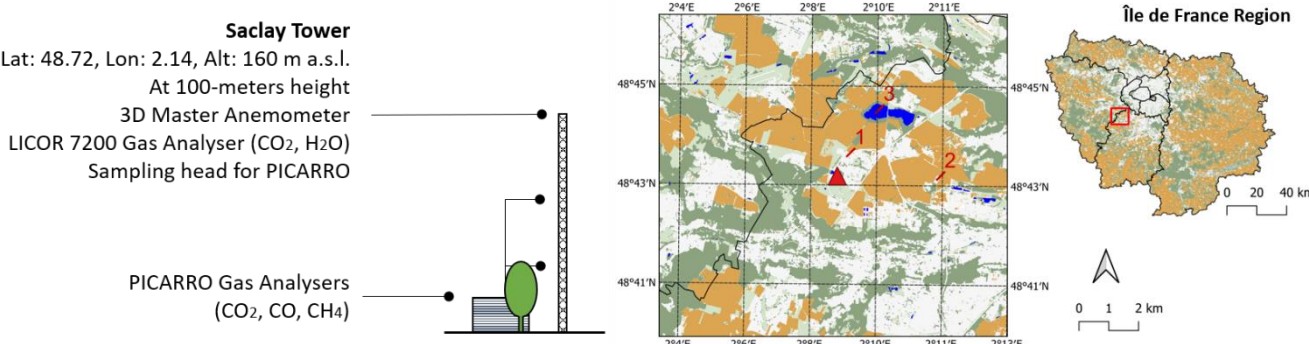

**Figure 1. Site diagram showing tower and acquisition house and tree for scale. On the right panel, site map and localisation in the region. Colours indicate land use: cropland (orange); grassland (light green); forest (dark green); water (light blue); white (urban). For reference (1) heating plant, (2) manure/composting plant, (3) lake.**

Since 2011 the site has been equipped with a high precision cavity ring down spectroscopy (CRDS) gas analysers ($CO_2$, CO, $CH_4$, G2401; Picarro Inc., Santa Clara, CA, USA) with a varying time response of a few seconds. The analyser is placed in a

ground level hut, connected to 3 sampling lines, 12.7 mm of diameter, collecting air at the 3 different heights of the tower (15, 60, 100 meters above ground level) alternatively every 10 minutes. Since 2017, a second multi-gas analyser is measuring continuously through a parallel sampling line connected to the top of the tower (100 meters above ground level).

The flow rate through the sampling lines is set around 12 L min$^{-1}$, but with no control. At the bottom of all lines connected to the CRDS analyzer the air is dried with Nafion (PermaPure, model MD-070-144S-4). The CRDS gas analysers were following

the ICOS calibration procedure aiming to a precision higher than 50, 1 and 2 ppb for $CO_2$, CO and $CH_4$ (ICOS RI, 2020). From June to October 2023, we setup a full Eddy Covariance system at 100 m, consisting of a closed-path infrared (IR) gas analyser (LI-7200; Li-Cor Inc., Lincoln, NE, USA), a 0.7 m heated tube with a flow rate set to 15 L min$^{-1}$ and a three-dimensional sonic anemometer (Gill WindMaster; Gill Instruments Ltd, Lymington, Hampshire, UK). The tower is also equipped with pressure (Vaisala PTB200), humidity and temperature sensors (Vaisala HMP155) at 1.5, 60 and 100 m.

Half-hourly average dry $CO_2$ mixing ratio showed a high degree of comparability between instruments ($R^2$ 0.97) and no bias (slope=1) (Figure 2). Nonetheless we found an offset of 7.25 ppm and an average drift of -11 ppm yr$^{-1}$, which has no impact on eddy covariance flux.





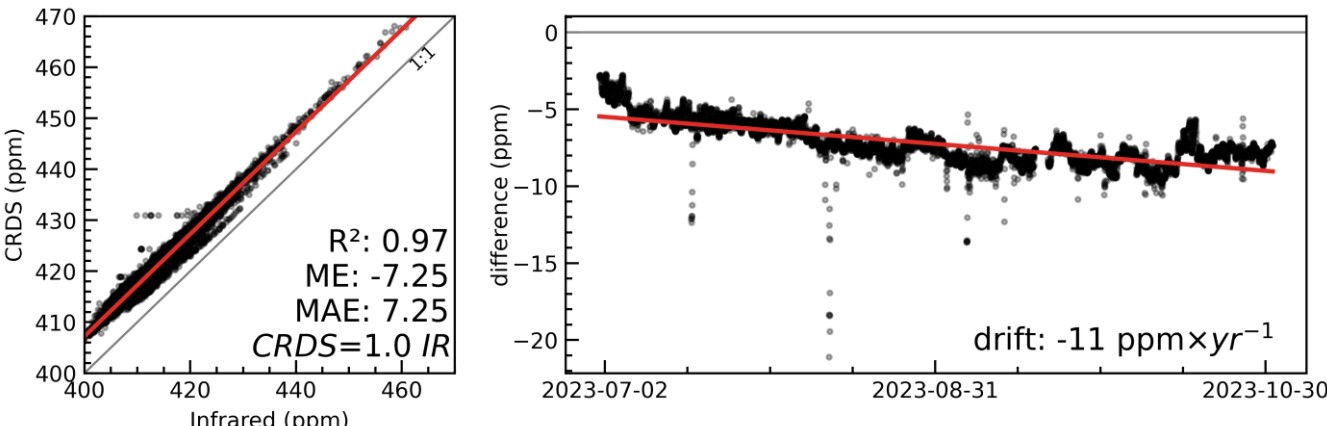

**Figure 2. Comparison between dry $CO_2$ mixing ratio measured by the IR (LI-7200) and the CRDS (PICARRO G2401) analysers.**
**Left panel: scatter plot. Right panel: mixing ratio difference (IR-CRDS) as a function of time. Dots are observations, red line is**
**linear fit and grey line is 1:1 line. The correlation coefficient ($R^2$), the mean error (ME, ppm), the mean absolute error (MAE, ppm),**
**the linear fit and drift.**

Data was not available for most of November due to maintenance in the instruments. The IR (LI-7200) was setup up

unconventionally with the analyser horizontally and the sampling tube vertically with a U-shaped head and a rain cap turned

downwards. The choice was made based on the safety for maintenance on top of the tower. Unfortunately, the IR malfunctioned

during the measurement campaign and prevented to have all analysers running at the same time for the whole period.

## 2.2. Data processing

The mass balance equation to compute surface flux includes three terms: storage; advection; and turbulent transport (Foken et

al. 2012). For the scope and goals of this work, vertical and horizontal advection were considered negligible, assuming dynamic

horizontal homogeneity of the surface. Vertical component of the wind ($w$) and the mixing ratio of a scalar $s$ ($\chi_s$) can then be

used to calculate flux at the surface $F_{eco}$ based on the turbulent covariance ($\overline{w'\chi_s'}$) measured at a certain height ($h_m$), and the

storage term ($\int_0^{h_m} \frac{\partial \overline{\chi_s}}{\partial t} dz$, where $t$ is time and $z$ is the height). Here overbars stand for time averaging. The surface flux $F_{eco}$

($\mu$mol m$^{-2}$ s$^{-1}$) can be then expressed as a function of the molar volume of dry air $V_a = \frac{RT_a}{P-e}$, where P is atmospheric pressure

(Pa), $e$ is vapour pressure (Pa), $R$ is the ideal gas law constant (8.31 J kg$^{-1}$ K$^{-1}$) and $T_a$ is air temperature (K):

$$F_{eco} = \left( \int_0^{h_m} V_a^{-1} \frac{\partial \overline{\chi_s}}{\partial t} dz + V_a^{-1} \overline{w'\chi_s'} \right) \tag{1}$$

### 2.2.1. Storage flux computation

The storage flux was computed as in Aubinet et al. (2005) as the derivative over time of scalar $s$ contained in the column below

the measurement height (100 m):



$$ST_s = \int_0^{h_m} V_a^{-1} \frac{\partial \overline{\chi_s}}{\partial t} \, dz \sim \frac{\Delta \sum_1^3 V_{a_i}^{-1} \overline{\chi_{s_i}} \Delta z_i}{\Delta t} \tag{2}$$

Where $\Delta t$ is 30 min, index $i$ stands for the three layers (0-15, 15-60, 60-100, and $\Delta z_i$ is the layer depth. The scalars $CO_2$, $CO$ and $CH_4$ were measured at 15, 60 and 100 m with the CRDS analysers. The dry air volume ratio was computed at each height based on measured air relative humidity and temperature. The storage was calculated using the three levels measurements done by the same instrument alternating between the three heights by periods of 10 minutes. The 10-minutes-average measurements were linearly interpolated. The 30-min average was computed and the time derivative calculated at that time-step.

**2.2.2. Turbulent flux calculation**

The turbulent flux was calculated based on a covariance, thus the name of the method Eddy Covariance (EC). Pre-processing is required, and was done using EddyPro 7.0.9, applying de-spiking (Mauder et al., 2013), covariance maximization for time lag, and double rotation (Wilczak et al., 2001). Time lag relates to the delay from sampling and measurement, and the maximization can lose reliability under noisy measurements (Langford et al. 2015). Typically, a default value and bounds are set individually for each gas and gas analyser. If an optimal value falls within the bounds, it is retained; otherwise, the default is chosen. For the LICOR 7200 analyser the lag time was set to $0.09 \pm 0.35$ s based on tube dimensions and flow rate. For the PICARRO analysers, that had a 100 m line, the lag time was set to $60 \pm 2$ s based on comparison with the LICOR 7200 $CO_2$ concentration. This lag time is compatible with flow rate ~12.6 L min$^{-1}$. The $\pm 2$ s tolerance was included to account for the uncertainty over the precise travel time and possible seasonal changes linked to air viscosity dependency to temperature and filter dirtiness.

In addition to the standard EC calculated by EddyPro 7.0.9, we used a second flux processing method based on discrete wavelet transform (Coimbra et al., 2023) (Appendix A). This method decomposes a time series ($x$) into sub-series ($\tilde{x}$), each defined in a given frequency domain $j$ :

$$x(t) = \sum_{j=0}^{J} \tilde{x}(t,j) \tag{3}$$

Where j is the scale level corresponding to a given frequency $f_j = s_0^{-1} 2^{-j\delta_j}$, for $j = 0, 1, \dots J$, where $s_0$ is the sampling rate (0.1 s in this study) and $\delta_j$ the frequency resolution (1 for discrete wavelets). When using discrete wavelets, since frequencies are independent (Coimbra et al., 2023, appendix A), we can calculate the covariance of $w$ and $\chi_s$ in each frequency band $\overline{\tilde{w}\tilde{\chi}_s}(j)$ by a simple multiplication of the decomposed sub-series $\tilde{w}(t,j)$ and $\tilde{\chi}_s(t,j)$:

$$\overline{\tilde{w}\tilde{\chi}_s}(j) = \overline{\tilde{w}(t,j) \times \tilde{\chi}_s(t,j)} = \int_{j-1}^{j} Co_{w\chi_s}(f) df \tag{4}$$

For continuous wavelets a different formula (Farge, 1992; Torrence and Compo, 1998; Farge and Schneider, 2001) or an empirical wavelet-specific-correction factor (Coimbra et al., 2023) should be applied. We note in eq. (4) that the frequency



resolved covariance $\overline{\widetilde{w}\widetilde{\chi}_s}(j)$ is also the integration of the cospectrum of $Co_{w\chi_s}(f)$ in the frequency band j-1 to j, which can be computed by EddyPro. The sum of the wavelet-decomposed covariance then yields the covariance:

$$\overline{w'\chi_s'} \approx \sum_{j=1..J} \overline{\widetilde{w}\widetilde{\chi}_s}(j) \tag{5}$$

A detailed description of the wavelet method, the wavelet transform and the corresponding flux data processing can be found in Appendix A1 and in (Coimbra et al., 2023). In this study, we used discrete Daubechies ($k=6$) wavelet (Daubechies, 1988) making sure the cone of influence was larger than the period selected. All fluxes were averaged every 30 minutes integrating up to the closest available period ($6.1\times10^{-4}$ Hz or 27 min). Despiking (Mauder et al., 2013) was used on each frequency-decomposed sub-series ($\tilde{x}$) to eliminate any unrealistic values identified and replaced them using a linear interpolation. For

EC flux calculation, the slow-response analyser (PICARRO) was resampled to 10 Hz to synchronize with the sonic anemometer sampling rate. This was achieved by repeating each measured value until it changed.

### 2.2.3 Quality flags and stability classes

Quality flags were assigned using the standard 0-1-2 flag system from FLUXNET (Mauder and Foken, 2011), involving tests for stationarity and fully developed turbulence (Foken and Wichura, 1996). Stationarity is essential to equate ensemble and

time averages, as turbulent fluctuation is formally defined as a deviation from the former rather than the latter. Standard eddy covariance (EC) cannot be used for non-stationary events, but wavelet decomposed series are stationary in each scale eliminating the need to flag out these data.

The stationarity test (STA) measures the absolute relative deviation between 5 and 30-minute covariances, while the turbulence test (ITC) assesses the deviation between measured and modelled integral turbulent characteristics. Data is considered of

quality high ($< 30\,\%$), medium ($30 – 100\,\%$), or low ($>100\,\%$), based on deviation percentages for each test (worst applicable result prevails). A detailed description of the quality flags can be found in Foken and Wichura (1996).

Stability classes were defined using the stability parameter $\zeta = (z - d) / L$, where $z$ is the measurement height, $d$ the zero-plane displacement height and $L$ the Obhukov length. We classified stability as: unstable ($\zeta < -0.2$); near neutral ($-0.2 > \zeta > 0.2$); stable ($\zeta > 0.2$).

**2.3. High frequency corrections on noisy measurements**

Instruments have measurement limitations which decreases their ability to produce a true value. Closed-path gas analysers require a gas sample to pass through a tube system including filters. Longer tube lengths typically result in increased time lag and reduced high-frequency signal. The signal degradation can be represented by a transfer function, $TF$, which attenuates the high frequency (Ibrom et al., 2007) of the true cospectrum of $w$ and a compound $s$:

$$fSp_{s,measured}(f) = fSp_{s,true}(f) \times TF \tag{6}$$





Where $f$ is the frequency (Hz), $Sp_s$ is the spectrum between $w$ and a scalar $s$. Note that we can considered the transfer function equal for the spectrum and cospectrum, as we neglect the $w$ transfer function and spatial sensor separation for the case of this tall tower (Massman, 2000). We assume the true covariance can be estimated by multiplying the measured covariance by a correction factor, $CF$:

$$\overline{w'\chi'_s}_{true} = CF \times \overline{w'\chi'_s}_{measured} \tag{7}$$

Acknowledging that the covariance is the integral over all frequencies of the cospectra, the correction factor CF can be

calculated from the transfer function TF and a true cospectrum, which is usually taken to be $wT_s$ (where $T_s$ is the ultrasonic temperature). Indeed, we assume similarity of scalars in the atmospheric boundary-layer, and use the (co)spectrum of $T_s$ as a proxy of unattenuated cospectrum, as $T_s$ is collocated to $w$ (Ibrom et al., 2007). This yields for $CF$:

$$CF = \frac{\int Co_{wT_s}(f)df}{\int Co_{wT_s}(f) \times TF(f|f_c)df} \tag{8}$$

Where TF can be calculated in different forms and can account for both low or high-frequency attenuation. Experimental methods are recommended for high-frequency spectral correction (Ibrom et al., 2007; Fratini et al., 2012). We can approximate

an empirical TF, explained further down, using a first-order system, as the product of a transfer function $H$ accounting for a first-order filter's time constant, $\tau_c$, representing the system response time (s), and a transfer function $H_p$ accounting for a generic phase shift φ as (Massman, 2000):

$$TF = H \times H_p \tag{9}$$

$$H = \frac{1}{1 + (2\pi f \tau_c)^2} \tag{10}$$

$$H_p = cos\varphi - 2\pi f \tau_c sin\varphi \tag{11}$$

Note that the cut-off frequency, $f_c$ equals $(2\pi\tau_c)^{-1}$. Ideally $H$ would be the measured-to-true spectra ratio for the scalar of interest. However, only the measured spectrum is known and so eq. (10(10) is fitted using the sonic temperature $T_s$ as a proxy

of the unattenuated spectrum (Ibrom et al., 2007; Fratini et al, 2012; Peltola et al., 2021):

$$H = \frac{Sp_s(f)}{Sp_{s,true}(f)} \approx F_n \frac{Sp_s(f)/\sigma_s}{Sp_{T_s}(f)/\sigma_{T_s}} \tag{12}$$

Where $F_n$ is a normalisation factor to account for any inaccuracies in the variance.

Sometimes $TF = H$ is used and $H_p$ is not considered (Ibrom et al., 2007). However, not accounting for the phase shift (e.g.: using cross covariance maximisation for lag correction and solely $H$ for cospectra correction) can bias CF (Peltola et al., 2021). Fortunately, $H_p \approx 1/\sqrt{H}$ which leads to $TF = HH_p \sim \sqrt{H}$ (Peltola et al., 2021). In this work we use Fratini et al. (2012) where

$TF = \sqrt{H}$.





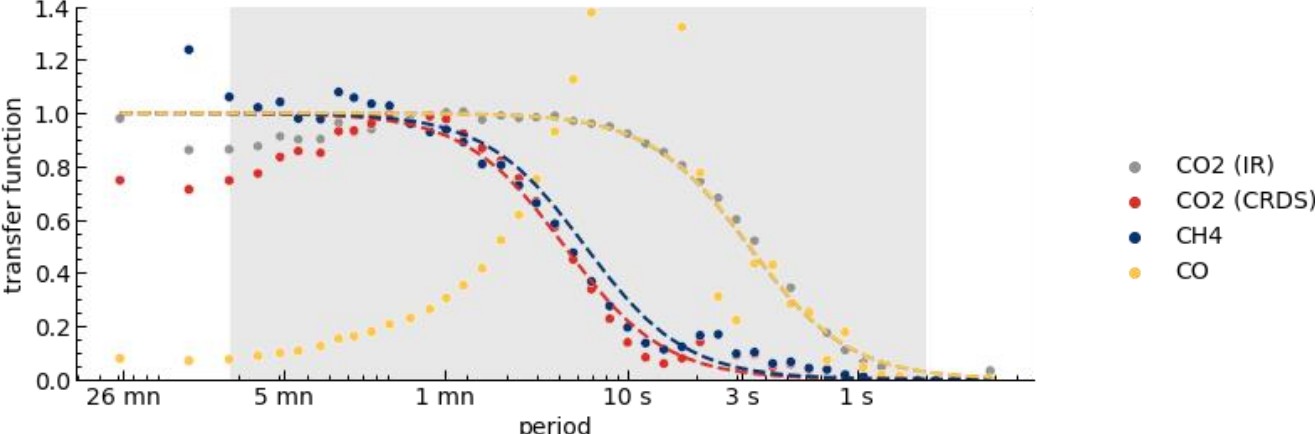

**Figure 3. Transfer functions $H$ (dotted lines) for each compound and analyser. The transfer function was fitted to the ratio of each compound's spectra to the sonic temperature spectra. All spectra are ensemble averaged taken from EddyPro outputs, filtered for significant flxues. Dots show the mean spectra per frequency band. The grey shaded area show the frequency range (2 - 0.0018 Hz) over which transfer functions were fitted.**

The spectra and cospectra calculation was performed using EddyPro 7.0.9, following Fratini et al. (2012) described here in equations 8-12. The transfer function $H$, accounting for the first-order filter's time constant $\tau_c$, was estimated for each analyser and each compound through a least square minimisation approach of the spectra (Figure 3).

From the $H$, $TF$ was computed as $\sqrt{H}$, and $CF$ was calculated with eq. (**8**). For both $TF$ optimisation and $CF$ calculation, only frequencies between 2 and 0.0018 Hz were used (see Figure 3). We assumed all compounds ($CO_2$, $CH_4$, CO) measured by CRDS (PICARRO G2401) suffered the same attenuation and used $CO_2$, the best-defined curve for all three analysers (Table 1). This assumption is grounded on the fact that measurements are done by the same instrument at the same acquisition rate sampled through the same line, and is backed by the proximity between $CH_4$ and $CO_2$ spectra, while the unexpected CO spectra can be explained by the CO signal noise due to the lower signal-to-noise ratio. Indeed, the noise was already larger than the signal at periods larger than 5 min (Figure 3). Similarly, the small step increase around 4 s for the CRDS analysers corresponds to the actual measurement interval.

**Table 1. The transfer function parameters for each instrument accounting for high frequencies attenuation. Here $TF = \left(1 + (2\pi f \tau_c)^2\right)^{-\frac{1}{2}}$, where $\tau_c$ is the first-order filter's time constant. The cut-off frequency, $f_c$ equals $(2\pi\tau_c)^{-1}$. $F_n$ is a normalisation factor. The optimized values correspond to optimisations as shown in Figure 3. The used values correspond to the optimised one except for the $CH_4$ and CO for which the $CO_2$ parameters are used instead. See equations (8-12) and text for details.**

| Instrument | Compound | Optimized | | | Used | |
| --- | --- | --- | --- | --- | --- | --- |
| | | $\tau_c$ (s) | $f_c$ (Hz) | $F_n$ (-) | $\tau_c$ (s) | $f_c$ (Hz) |
| IR | $CO_2$ | 0.5 | 0.34 | 1.12 | 0.5 | 0.34 |
| CRDS | $CO_2$ | 3.0 | 0.05 | 1.63 | 3.0 | 0.05 |
| CRDS | $CH_4$ | 2.4 | 0.06 | 1.01 | *3.0* | *0.05* |





| CRDS | CO | 0.5 | 0.34 | 3.49 | 3.0 | 0.05 |
|------|----|-----|------|------|-----|------|

### 2.4. Spatial tools

For an analysis of the fluxes' footprint as a function of wind direction, we used a backward Lagrangian stochastic particle
dispersion model (LPDM-B) for the footprint (Kljun et al. 2015), and computed vegetation indexes based on Sentinel 2
(ESA/Copernicus Data) and a French land use map (IGN, 2022).

Sentinel 2 data was collected using Google Earth Engine with a tool available at https://github.com/pedrohenriquecoimbra. To
monitor vegetation, we calculated the enhanced vegetation index (EVI) using the following equation:

$$EVI = G \frac{NIR - Red}{NIR + C_1 Red - C_2 Blue + L} \tag{13}$$

Where NIR, Red, and Blue are surface reflectances centred in the 842, 665 and 490 nm wavelengths, band B8, B4 and B2 in
Sentinel2, corrected for transfer through the atmosphere; L is the canopy background adjustment that addresses non-linear,
differential NIR and red radiant transfer through a canopy, and $C_1$, $C_2$ are coefficients, G is a gain factor. We adopted the same
coefficients as in the MODIS-EVI algorithm: L = 1, C1 = 6, C2 = 7.5, and G = 2.5. For all reflectance bands, we removed
clouds using the Sentinel-2 Cloud Masking, *s2cloudless*, also available in Google Earth Engine. We classified clouds if the
cloud probability was above 60 % and removed pixels considered as cloud shadows based on a threshold of 0.15 for near
infrared and a maximum distance of 1 km from cloud edges. We also removed 50 m around the mask assuming these pixels
may still be affected by the cloud shadowing.

### 2.5. Performance measurements

Comparisons between instruments were carried out using mean bias and absolute error, defined as:

$$Mean\ Error\ (bias) = \frac{1}{N} \sum_{n=1}^{N} (X_{a,n} - X_{b,n}) \tag{14}$$

$$Mean\ Absolute\ Error = \frac{1}{N} \sum_{n=1}^{N} |X_{a,n} - X_{b,n}| \tag{15}$$

Where *N* equals the amount of data, *X* is the variable measured with instrument *a* and *b* at a time *n*.
In figures, 95% confidence interval bands were calculated using the Seaborn module in Python. It uses a random sampling
with replacement strategy, bootstrapping, to construct a confidence interval (Dragicevic, 2016).

For linear fits, if not declared otherwise, the squared loss, also named ordinary least squares method, is used. The method
consists of minimizing the sum of the squares of the difference between the observed and predicted values. When robust or





Huber loss is mentioned, we use a linear fit which minimizes the squared loss for the samples where the absolute difference

between the observed, $y$, and predicted, $f(x)$, values is smaller than $\delta$ and the absolute loss, sum of the absolute difference, otherwise. This feature makes it less sensitive to outliers than the squared error.

$$
Huber\ loss = \begin{cases} \sum \frac{1}{2}(y_i - f(x_i))^2, & |y_i - f(x_i)| \leq \delta \\ \sum \delta(|y_i - f(x_i)| - \frac{1}{2}\delta), & otherwise \end{cases} \tag{16}
$$

By default, we chose arbitrarily $\delta = 5$, note that very low $\delta$ values may increase the number of values considered as outliers.

## 3. Results

### 3.1. Mixing ratios of CO₂, CO and CH₄

In Figure 4, diel pattern shows a peak in $\chi_{CO_2}$ during morning, 07:00 in July and moving towards 09:00 in October, and a clear valley around 15:00. The pattern disappears when moving towards winter months. CO and CH₄ both show a similar peak in the morning autumn, although less marked. Only CO shows an afternoon peak in September which is also the month with the clearer morning peaks for CH₄ and CO. Seasonally CO₂, CO and CH₄ mixing ratio are the highest in winter while H₂O is higher during summer. This difference may be explained by a larger biogenic CO₂ sink and H₂O source during daytime in

summer and a higher anthropogenic CO₂ emission in winter (heating on). The difference may also be explained by larger (smaller) boundary layer thickness during the summer (winter) which can effectively dilute (concentrate) the molecules emitted at the ground.

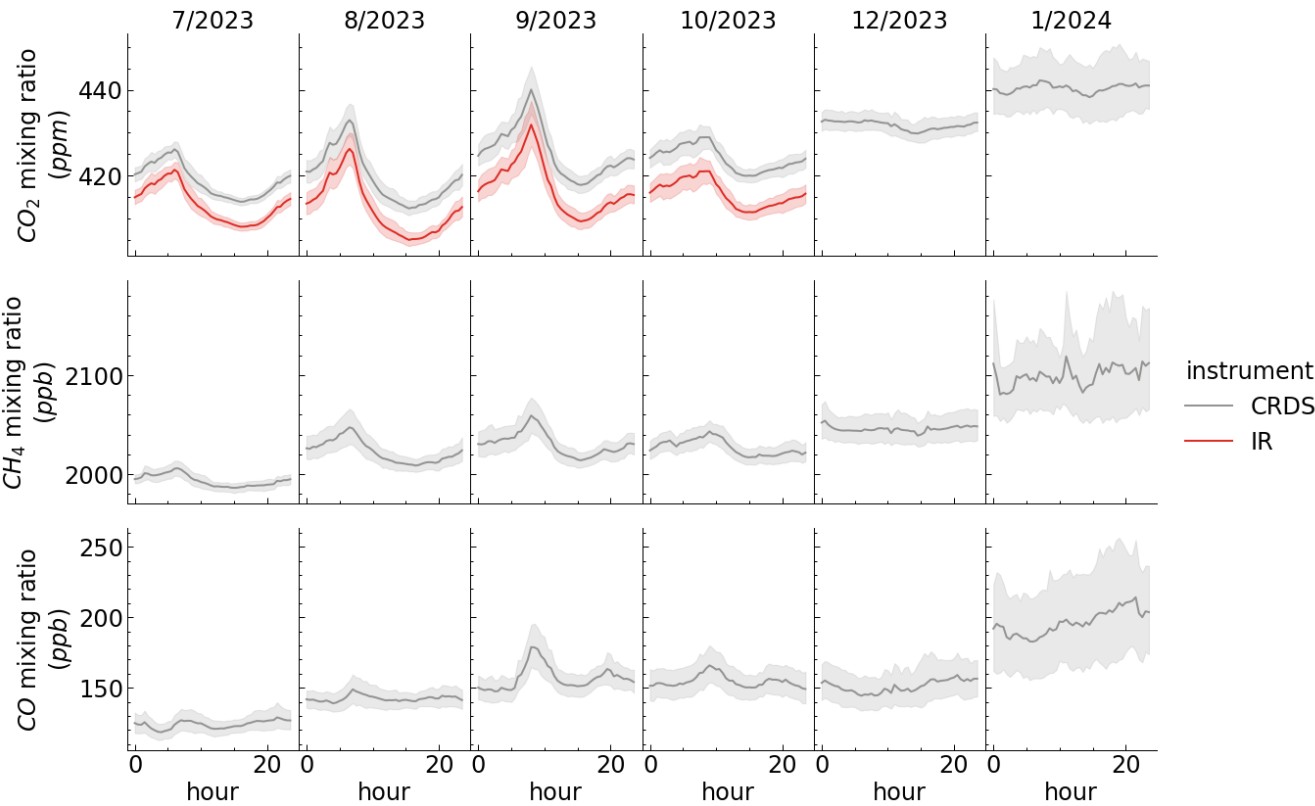

**Figure 4. Monthly dry mixing ratios diel pattern for all measured gases ($CO_2$, $H_2O$, $CH_4$ and CO) for IR (LI-7200) and CRDS (PICARRO G2401). Solid line indicates median and region shows 95% confidence interval.**

A look into how the mixing ratios vary with wind direction reveals some spatial patterns (Figure 5). During warmer months (July to October), Westwind $CO_2$ mixing ratio was smaller than the median value, while for $CH_4$ and CO we can notice a higher than the median value for Northeast sector, especially clear for CO. In colder months (December and January) all mixing ratios were higher (also seen in Figure 4), with North-Easterly winds (0-180°) showing larger mixing ratios than in other directions. A peak in mixing ratios is observed for all three gases for winds coming from around 20°N, the direction from the heating plant. Interestingly, a smaller peak can be seen on the Northwest, direction from the lake (100 m afar), bare soil fields (around 500 m afar) and a regional road roundabout (around 1 km afar).


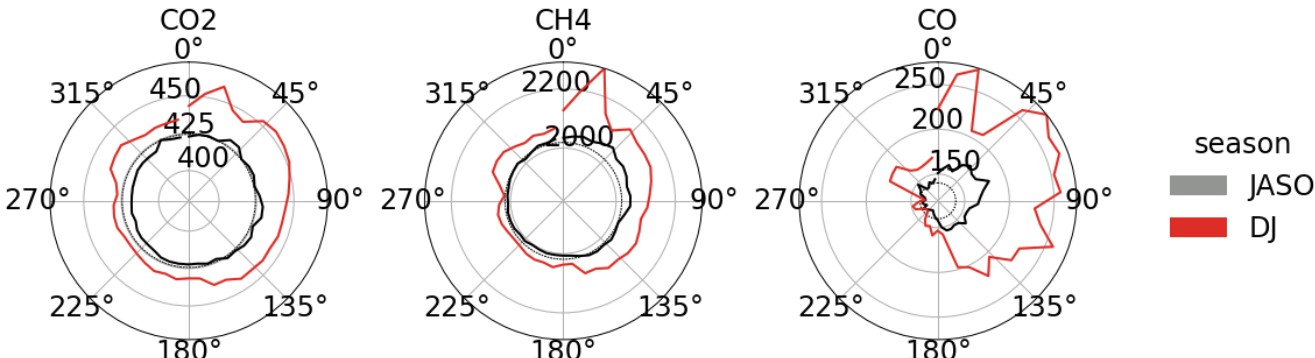

**Figure 5. Average dry mixing ratios by wind direction. Warmer months (July to October) in grey and colder months (December and January) in red. Extreme values in the left and right 0.1% tails were removed. See Figure S1 for monthly observations. $CO_2$ is in ppm while $CH_4$ and CO are in ppb.**

### 3.2. Footprint and stationarity

### 3.2.1. Footprint analysis

A characterization of the site's flux footprint (Figure 6) shows a heterogeneous landscape composition, comprising 25% urban, 23% agriculture, 21% forest, and 21% grassland areas. In the western part of the site (42% forest), there is a relatively dense woodland primarily featuring deciduous trees. To the south (41% grassland), the landscape includes a nearby golf club in the vicinity of the CEA campus. In all directions there are croplands, predominantly cultivated cereal crops (winter wheat, barley, maize) and oilseeds (rapeseed), typical of the region. In the northeast (45% urban), the landscape aligns with the location of the CEA campus which includes a heating plant aligned to 20° N. The 43.9 km² 80% footprint encompasses a national road (N118) and several regional roads (D306, D36, D128) with a weekly traffic of 60 thousand vehicles on average (in 2022 according to SIREDO). Water ponds have a small contribution for northwest to northeast sectors (2.3-2.6%). In these sectors two ponds are situated one approximately at 100 meters from the tower (northwest) and a second larger farther away (around 2.4 km northeast, visible in the map Figure 6.a).



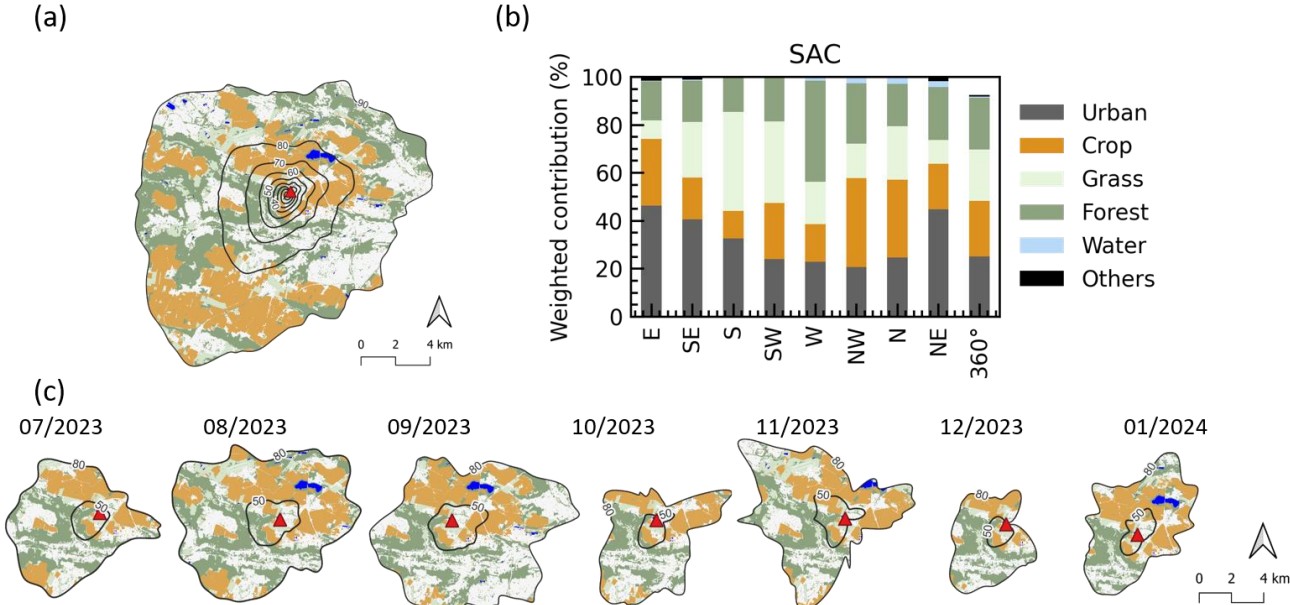

**Figure 6. Flux footprint by land use group. (a) Footprint for all the period, where lines indicate 10 to 90 (border) % level source area. (b) Contribution of each land use weighted by footprint density. (c) Monthly footprint, where line indicate 50 and 80 (border) % level source area. Note that for visual purposes urban is coloured as white-grey in the map. Footprints estimated using model in Kljun et al. (2015).**

In Figure 6.c, we can see that the monthly changes in composition and shape of the area contributing to the fluxes measured at the tower (the flux footprint). Some months have larger footprint (e.g. August and September) while others are narrower (e.g. December and January) related to changes in the dispersion conditions. This difference is explained by the largest occurrence of stable conditions during the summer which leads to larger footprints than during the winter that has mainly neutral conditions (shown by the stability ratio $z / L$) driven by stronger Winter winds, elevated friction velocity, and cloudy conditions (Figure 7). Note as well that the landscape is not homogeneous (Figure 6.b), and so wind direction can also change the effective profile of sources and sinks contributing to each compound flux measured at the tower. December, for instance, was the month with the least contribution from the most urban northeast sector.

The mixing layer height, and similarly the atmospheric boundary layer height, shows a clear diurnal and seasonal cycles (Figure 7). Warmer hours of the day and months show taller boundary layer heights implying a larger volume of developed layer in which the compounds can be diluted. During these warmer periods the conditions are often unstable ($z/L < 0.2$) and friction velocity is high ($> 0.4$ ms$^{-1}$). This indicates well-mixed layer and bigger eddy sizes. On the contrary, colder months (December and January) showed relatively flat diel pattern, mostly due to a shorter photoperiod, leading to a much lower boundary layer height. We also noted on-site fog was frequently observed during these periods. Concurrently, friction velocity increased on average during winter. We not that strong winds in neutral conditions and especially medium winds in stable conditions would





be favourable to horizontal advection. In the scope of the present work, however, it was not quantified due to a lack of
measurements (horizontal gradients of concentration and fluxes).

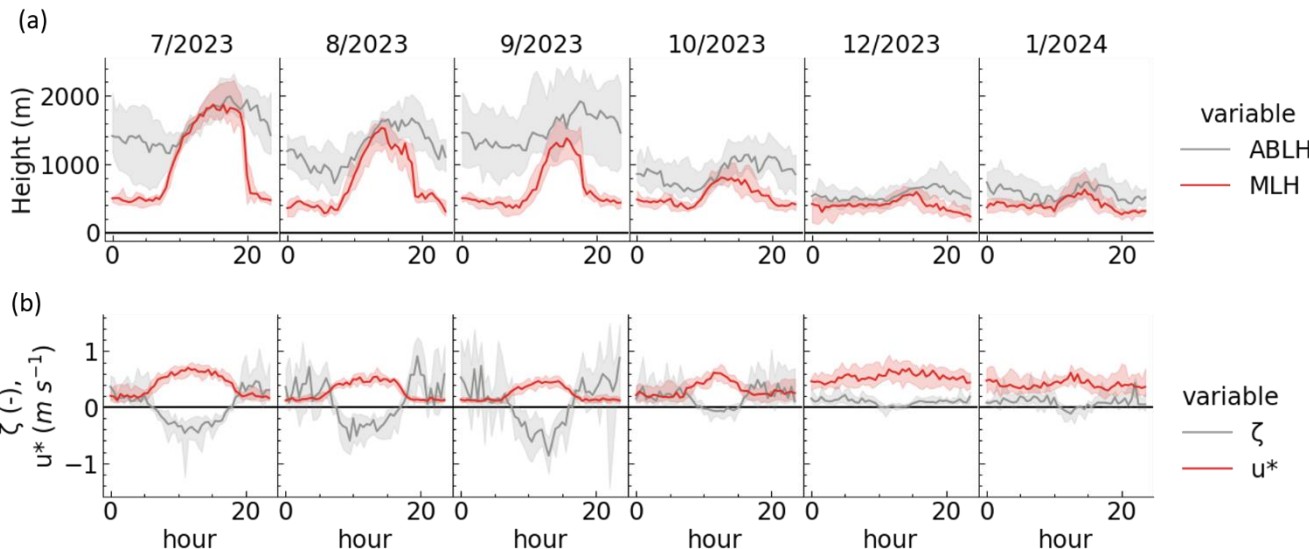

**Figure 7. Boundary layer conditions. (a) Heights of the atmospheric boundary layer (ABLH) and the mixing layer (MLH) measured by SIRTA in Palaiseau, 4.8 km away from the tower. Data available online (Kotthaus et al., 2023). (b) Stability parameter ($\zeta = (z - d) / L$) and friciton velocity ($u_*$) measured at the FR-Sac tower. Absolute values of $\zeta$ bigger than 2 were ignored.**

### 3.2.2. Stationarity and well-developed turbulence

Most of the data collected was under well-developed turbulence, 75% if only considering high-quality (flag 0) integral turbulence characteristics test (ITC), and 99 % including medium-quality (flag 1). Around half of the data (41 %), with an ITC flag 0 was also considered stationary (Stationarity flag 0), increasing to 81 % if we include flag 1 on both tests. The stationarity test is required for standard EC but not for wavelets, thus the use of the latter increases the data amount by 34 % in case only
high-quality observations are used and 55 % in case medium-quality data is included (Figure 8). This savings happens more often during the day, due to a higher coincidence of both flags during night. The percentage given are for the Licor (IR) fast analyser but are of the same order of magnitude for the PICARRO analyser (CRDS).





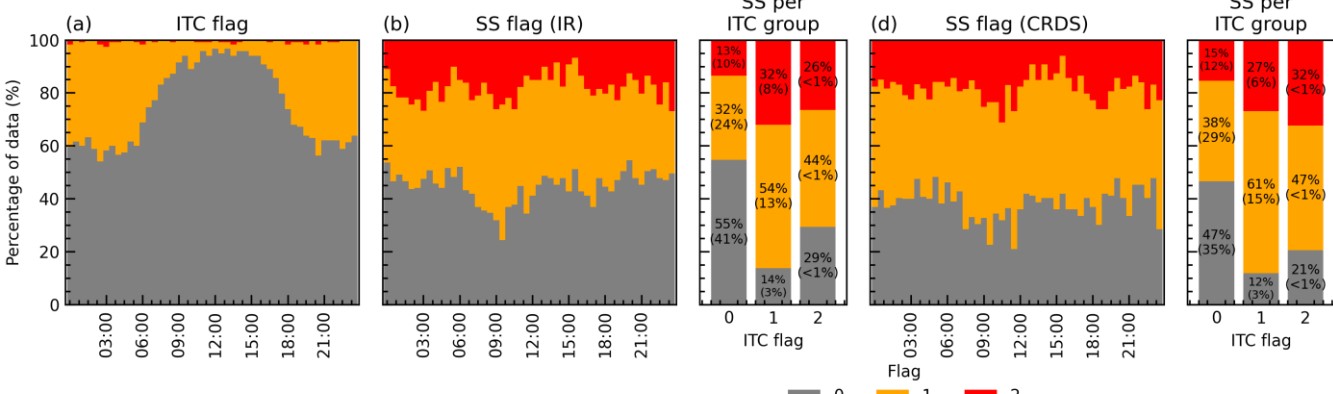

**Figure 8. Quality control flags for turbulence (ITC) and CO₂ stationarity (SS). Flags follow 0-1-2 system for high, medium and low-quality. Percentage of (a) turbulence flagged data by hour of the day. Stationarity flagged CO₂ data by hour of the day for the (b) the Licor instrument (IR) and (d) the PICARRO instrument (CRDS). Stationarity flag per ITC group are also given for the IR (b) and CRDS instrument (c). Percentages are summed to 100% in each group and over all data (in parentheses). See Figure S2 for stationary test for the three instruments.**

## 3.3. Comparison of CO₂ flux between slow and fast-response analysers

### 3.3.1. High frequency spectral correction

The CRDS analysers showed significantly more high-frequency attenuation of the flux than IR analysers (Table 2 and **Figure 9**), as expected due to the much longer sampling tube of the CRDS analyser (115 m) than the IR analyser (0.7 m), as well as the slower CRDS acquisition frequency (~3 s) compared to the IR (0.1 s). The difference was greater in (very) stable conditions, when higher frequencies contribute more to the flux, than on (very) unstable conditions (**Figure 9**). On (very) unstable conditions the contribution of low-frequencies to the flux increased as shown by the fact that none of the ogive levelled to 1 towards 30 minutes integration time (ogive slope > 0). Surprisingly CO (measured by CRDS) showed an atypical curve with stronger contribution from high frequencies, which after analysis was attributable to noise from this less sensitive instrument.





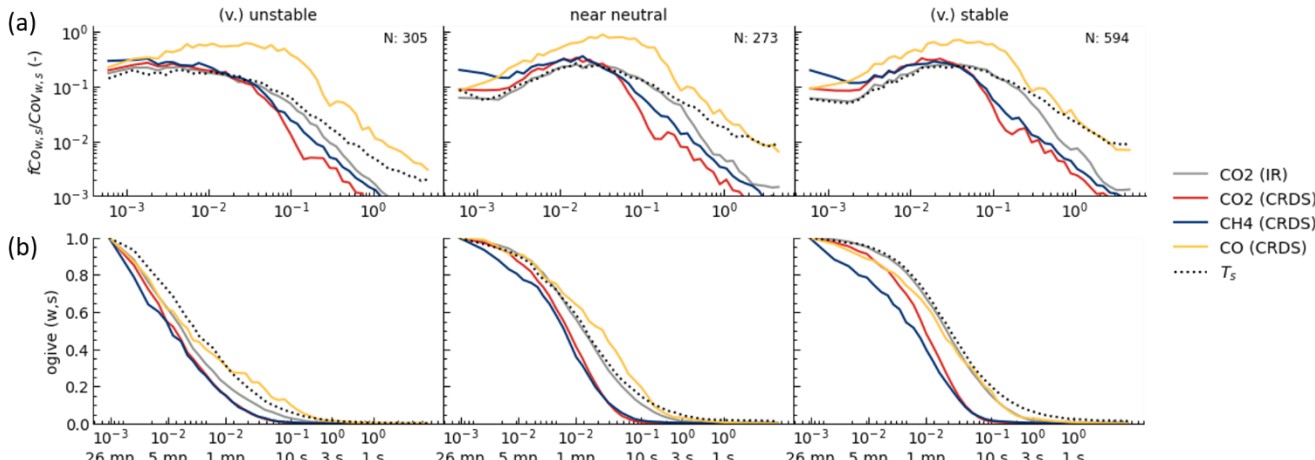

**Figure 9. Normalised cospectra (a) and ogives (b) of w and $CO_2$, $CH_4$, CO covariances, for gases measured by IR and CRDS and the reference sonic temperature, $T_s$. Median values from July to October 2023 grouped by stability classes: $\zeta < -0.2$ (unstable); $-0.2 > \zeta > 0.2$ (near neutral); $\zeta > 0.2$ (stable). N indicates the amount of half hourly data in each class.**

The high-frequency attenuation varied from 3 to 7% for the fast instrument (IR), while for the CRDS instruments sampling at 100 m, it ranged from 11 to 19% (Table 2). We can expect larger corrections on stable conditions, characterized by a larger contribution of high frequencies to the flux, as observed for IR, contrarily CRDS shows a decrease compared to near-neutral. It is worth noting that despite the 10 Hz acquisition frequency and 100 m height, the attenuation of the IR instrument was non-negligible. Additionally, the time response of the slow CRDS analysers, estimated based on the transfer function (3.62 s), matches the acquisition frequency (ranging between 3 s and 4 s), but it also matches the expected attenuation for a long tube (Figure 16).

**Table 2. Percentage high frequency corrections of the $CO_2$, $CH_4$ and CO fluxes per stability class for each instrument. Note we used the $CO_2$ transfer function for all compounds in the CRDS assuming the damping in the sampling line was dominant attenuating process.**

| Instrument (compound) | Stability class | | |
|---|---|---|---|
| | (very) unstable | near-neutral | (very) stable |
| IR ($CO_2$) | 2.7% | 5.5% | 6.6% |
| CRDS ($CO_2$, $CH_4$, CO) | 11% | 19% | 17% |

### 3.3.2. Comparing $CO_2$ flux measured by slow and fast-response analysers

The $CO_2$ fluxes computed from the IR (LI-7200) and the CRDS (PICARRO) analysers were well correlated with an underestimation of 13% of the CRDS for uncorrected fluxes that was diminished to 3% after high frequency corrections (Figure 10). High-frequency correction decreased the bias, ME, by 0.04 $\mu molm^{-2}s^{-1}$ with no effect for absolute error (MAE) or the correlation coefficient ($R^2$). There was a moreover a tendency of the CRDS corrected fluxes to slightly underestimate the $CO_2$ fluxes under stable conditions (Figure S3).





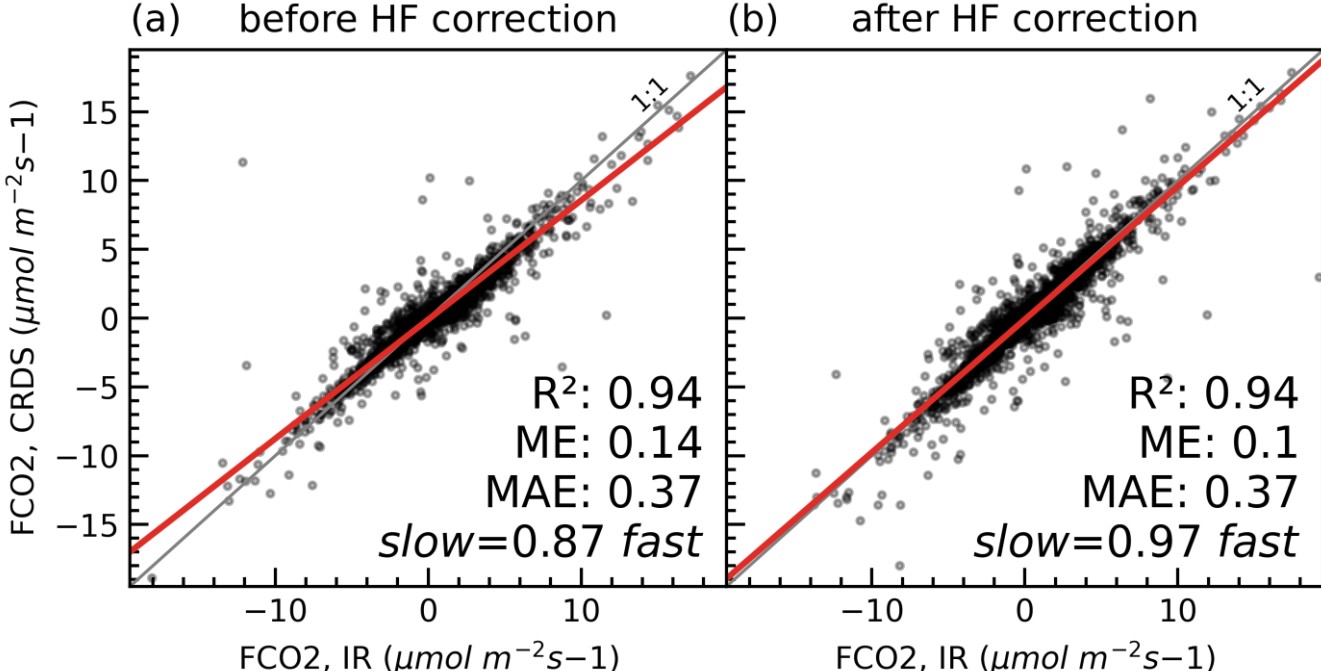

**Figure 10. Comparison of CO₂ flux computed with the IR (LI-7200) and the CRDS (PICARRO G2401) analysers, (a) before and (b) after high frequency losses corrections. Dots are observations, red line is a robust linear relation and grey line is the 1:1 line. The correlation coefficient (R²), the mean error (ME, µmolm⁻²s⁻¹), the mean absolute error (MAE, µmolm⁻²s⁻¹). Statistics are calculated ignoring outliers from robust linear regression.**

### 3.4. Surface flux dynamics

### 3.4.1. Turbulent fluxes

We observed a well-defined summer pattern for the $CO_2$ flux with emissions during the night and sequestration during the day (Figure 11). From summer to winter the sink shortens in time and decreases in magnitude up to the point that during winter, the site behaves on average as a source all along the day. We note that the concentration morning peak observed in Figure 4 does not correspond to a peak in the flux. Following the seasonal pattern of $CO_2$, the evapotranspiration (as shown by the latent heat flux) decreased towards colder months. The similarity between $CO_2$ and $H_2O$ trends is a good indication that photosynthetic activity was slowing down. Indeed, in September several crops were senescent or harvested and the deciduous trees in the surrounding started to lose their leaves as shown by the EVI maps (Figure S5).

The $CH_4$ fluxes showed a quite marked daily pattern from July to September with higher emissions in the morning than in the afternoon. Seasonally the emissions in January were a factor of 10 larger than in the previous months. Looking at the CO flux, we see a marked increase in November and January but not in December, despites similar temperatures and traffic. In January winds were relatively well distributed while in December the most urban NE sector was rarely in the footprint, which may explain the difference between the three months.



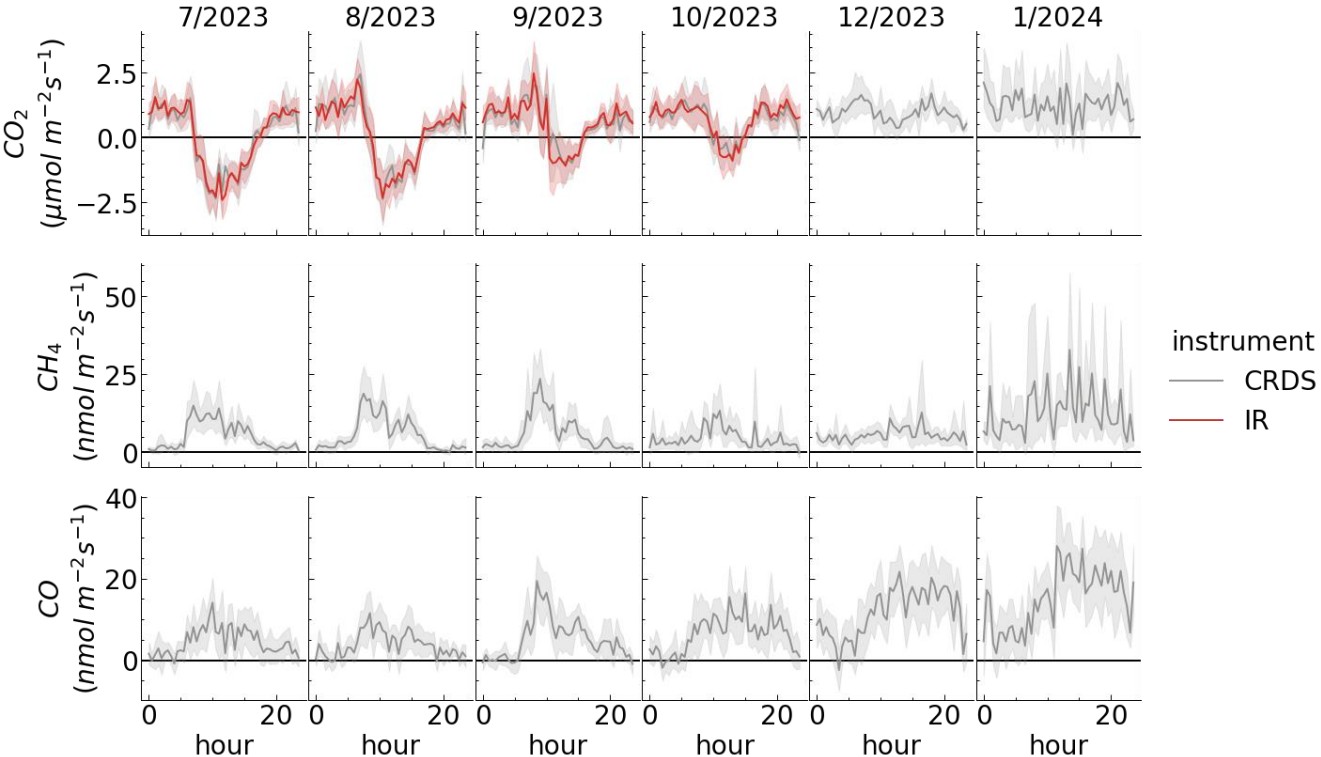

**Figure 11. Monthly mean turbulent fluxes diel pattern of CO₂, CH₄ and CO for IR (LI-7200, only CO₂) and CRDS (PICARRO**
**G2401) gas analysers. Fluxes showed after spectral correction. Data points falling within the extreme 1% tail of the distribution**
**were removed.**

### 3.4.2. Storage and surface fluxes

The $CO_2$ storage fluxes were of the same order of magnitude as the turbulent fluxes (Figure 11) and showed diurnal patterns

with positive values at night, early peaks of negative values during sunset, and a following increase towards positive values

throughout the day (Figure 12).

Overall the storage fluxes led to increased night-time emissions and daytime absorption of $CO_2$ and $CH_4$ surface fluxes

compared to turbulent fluxes measured at 100 m. In warmer months, when storage was significant, we observe the appearance

of destocking (negative storage term) at the same time as the rise in mixing layer height (Figure 7). Both started at early hours

when the surface heated by the sun sets up instability leading to the formation of turbulence. During these early events, nearly

all the ecosystem flux was measured through $CO_2$ storage flux. Over the months we observe a decrease in the $CO_2$ storage

term as the atmospheric stratification progressively becomes mainly neutral and the boundary layer height stayed unchanged

throughout the day. We also note a decrease in $CH_4$ over time, however, in January we can see negative values in the middle

of the day, also observed for CO and $CO_2$. CO showed little storage during the vacation months (June and July) or in December

with wind coming from the vegetated sector. CO storage increased in September and October during rush hours.





Figure 12. **Monthly mean storage fluxes diel pattern of CO₂, CH₄ and CO computed using mixing ratios measured by the CRDS (PICARRO G2401) gas analyser on three levels (15, 60 and 100 m). Data points falling within the extreme 1% tail of the distribution were removed.**

**3.5. A look into the flux spatial heterogeneity**

Fluxes categorized by wind direction and separated for day and night reveal spatial patterns (Figure 13). During daytime in warmer months (January to October), most directions exhibited a $CO_2$ sink. The most pronounced $CO_2$ sink was observed from the west, in the direction of the forest, where some emissions of $CH_4$ were observed, but notably smaller emissions of CO. Colder months revealed stronger southeast $CO_2$ emissions, echoed in $CH_4$ and CO fluxes. In the northeast sector, CO emissions were strong, increasing in colder months, possibly due to the alignment with the national road N118, ~50°. Peaks in CO emissions during nights align with directions with higher emissions during the day and may be turbulent fluxes from particularly windy nights. Additionally, the wind direction spanning 10-45°, direction of the local heating plant, exhibited $CO_2$ and $CH_4$ fluxes significantly higher than those observed in other directions, during both daytime and night-time periods.

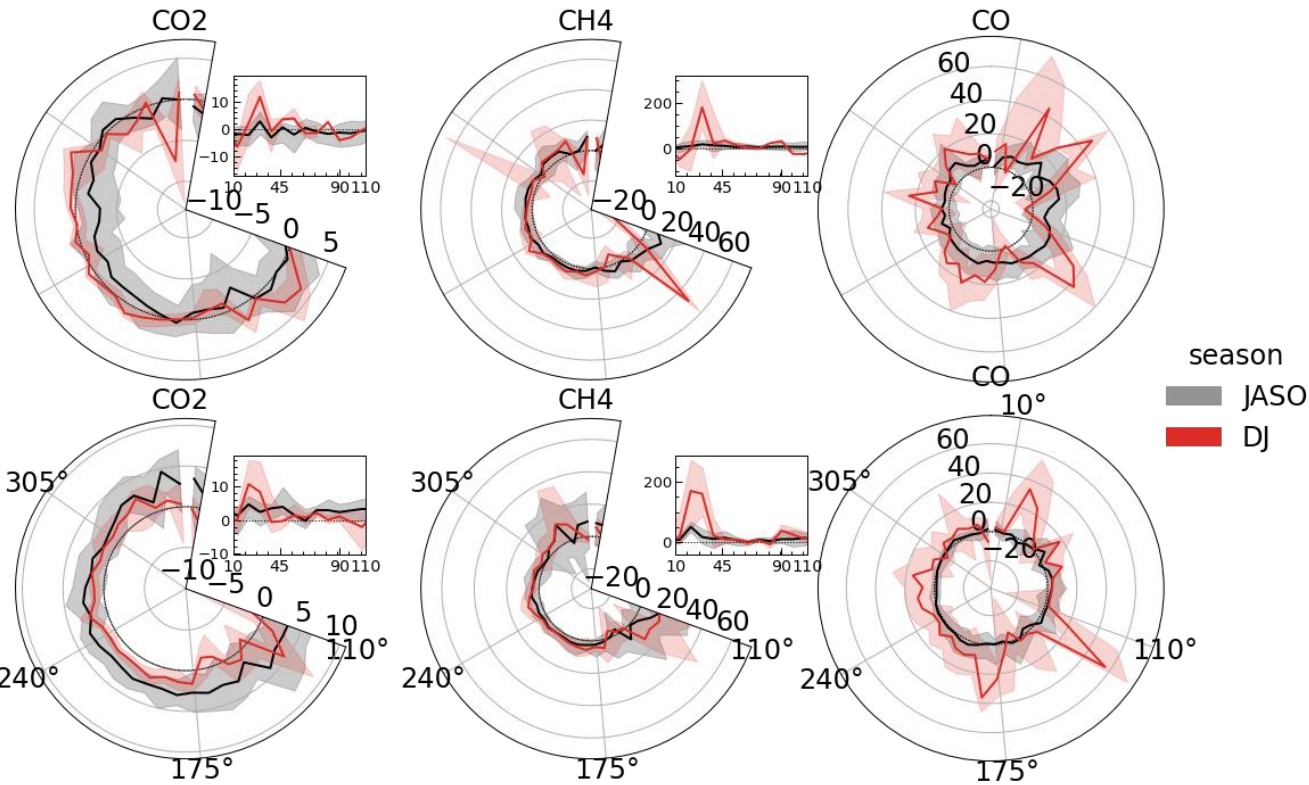

**Figure 13. Surface fluxes by wind direction, daytime in the top panel and night-time in the bottom panel. Surface fluxes includes**
**turbulent and storage terms. The values presented are the median with the interquartile range, 0.1% extreme values were removed.**
**Wind directions bins with less than 10 observations were added to the next bin clockwise. $CO_2$ fluxes are in µmol m$^{-2}$ s$^{-1}$ and fluxes**
**of $CH_4$ and CO are in nmol m$^{-2}$ s$^{-1}$. Note for $CO_2$ and $CH_4$, 10-110° were plotted separately for visual purposes. See Figure S4 for**
**monthly values.**

The NW $CO_2$ flux also showed a shift from source in warmer months to a small sink during winter. This shift is expected when
we consider the crop field on this wind direction was bare soil in July (Figure 14) and green in January (Figure 15). The

greenness can be identified by a higher leaf density, which the Enhanced Vegetation Index (EVI) serve as a proxy.

In July, west wind fluxes show a recognizable $CO_2$ diel pattern with carbon sequestration during the day when wind comes

from the vegetated direction (W and SW) (Figure 14.b). The high EVI indicates that the forest and grassland had fully green

leaves during this month. On the 17$^{th}$ July, we can spot a positive $CO_2$ peak during the end of the day coming from the W

direction, not found in $CH_4$ nor CO. The peak is atypical at this magnitude for a forest ecosystem and may be a signal from

the road or other activity. $CH_4$ was found to be more elevated for W winds. CO did not show a marked pattern when wind

came from vegetated direction. For the period when wind came from the CEA campus (10-100°) the three gases showed very

similar patterns, with a peak in the middle of the day.



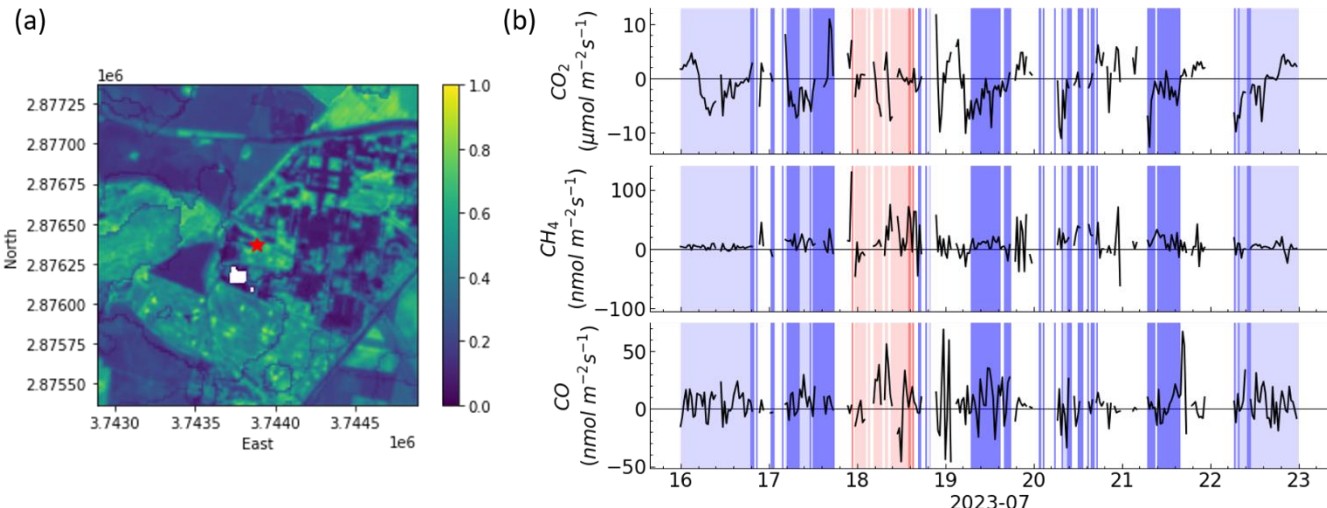

**Figure 14. Zoom into a summer week. (a) EVI mean mosaic from July (2nd, 7th, 12th, 14th, 17th, 24th) using Sentinel2 data. (b) Surface fluxes, turbulent and storage term, with background colours per wind directions, W in blue, SW in light blue, 10-30° in red, 30-100° in light red.**

In January, when the heating plant was operational and the wind came from its direction, we can distinctly observe the contribution of the heating plant to the $CO_2$ and $CH_4$ fluxes (Figure 15). CO fluxes were also higher when wind came from the heating plant but the difference compared to the vegetated wind direction was less striking. The mean (and 90th percentile) for





$CO_2$, $CH_4$ and CO fluxes from the NE were 8.2 (25) µmol m$^{-2}$ s$^{-1}$, 214 (801) nmol m$^{-2}$ s$^{-1}$, and 27 (71) nmol m$^{-2}$ s$^{-1}$, respectively. While for W and SW values were 1 (2.3), 4.8 (12) and 12 (40) µmol m$^{-2}$ s$^{-1}$ smaller.

Apart from the fluxes coming from the heating plant, Wintertime fluxes were notably smaller compared to summer (Figure 14), as expected by the lower biological activity during that period. The green areas exhibited lower EVI, indicating the loss of leaves during this season.

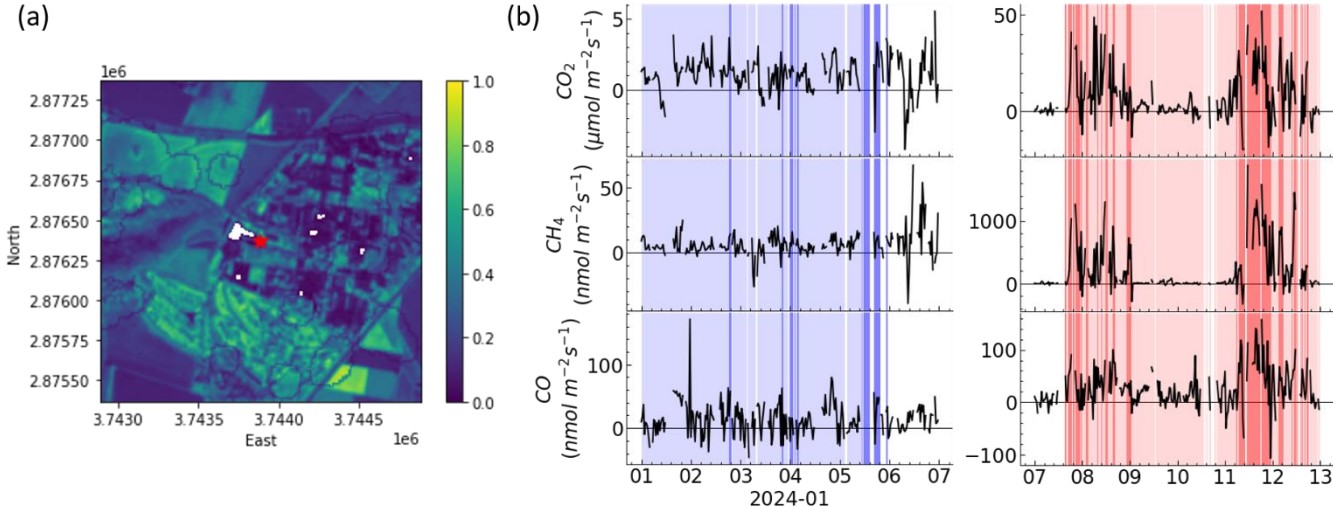

**Figure 15. Zoom into a winter week. (a) EVI mean mosaic from January (5$^{th}$, 18$^{th}$, 20$^{th}$) using Sentinel2 data. (b) Surface fluxes, turbulent and storage term, with background colours per wind directions, W in blue, SW in light blue, 10-30° in red, 30-100° in light red.**

## 4. Discussion

### 4.1. Challenges of measuring on a tall tower with slow-response analysers

#### 4.1.1. High frequency losses corrections on the atmospheric tower configuration

Our findings revealed that an ICOS atmospheric tower configuration, utilizing a CRDS gas analyser with an acquisition frequency of approximately 0.3 Hz and a tube length of 100 m, exhibited a high-frequency loss correction of approximately 20%. This correction was around three times more than that of the conventional ecosystem flux measurement setup, which employed an IR gas analyser with a 10 Hz acquisition frequency and a tube length of 0.7 m, positioned at the top of the tower. The observed transfer function (TF) for the CRDS setup closely matched the theoretical attenuation expected, as depicted in Figure 16. Indeed, the tube and sensor attenuation together lead to a first-order time constant around 3s as we observed for the CRDS setup (Table 1). This outcome suggests that even with a faster measurement system or a smaller tube attenuation, only a limited reduction of the attenuation can be expected. In order to substantially decrease the high frequency attenuation of the flux, both an increase of the acquisition frequency and a decrease in tube attenuation (decrease in tube length or increase in flow rate) would be required.





We note that since the tube attenuation is higher when flow inside the tube is laminar (Lenschow and Raupach, 1991), ensuring a Reynolds number larger than ~2300 is key to minimize attenuation. We can define $Re = \frac{2Q}{\pi r v}$, where Re is Reynolds number

(-), $r$ is tube radius (m), Q is volumetric flow rate ($m^3 s^{-1}$), and $v$ is the kinematic viscosity of air. We find that for the tube in place with 9.5 mm of internal diameter, pumping ~14-17 L min$^{-1}$ is necessary to achieve a turbulent flow. Under these conditions, the cut-off frequency would increase to more than 0.6 Hz, but the pressure would also drop from -6 mbar to -47 mbar.

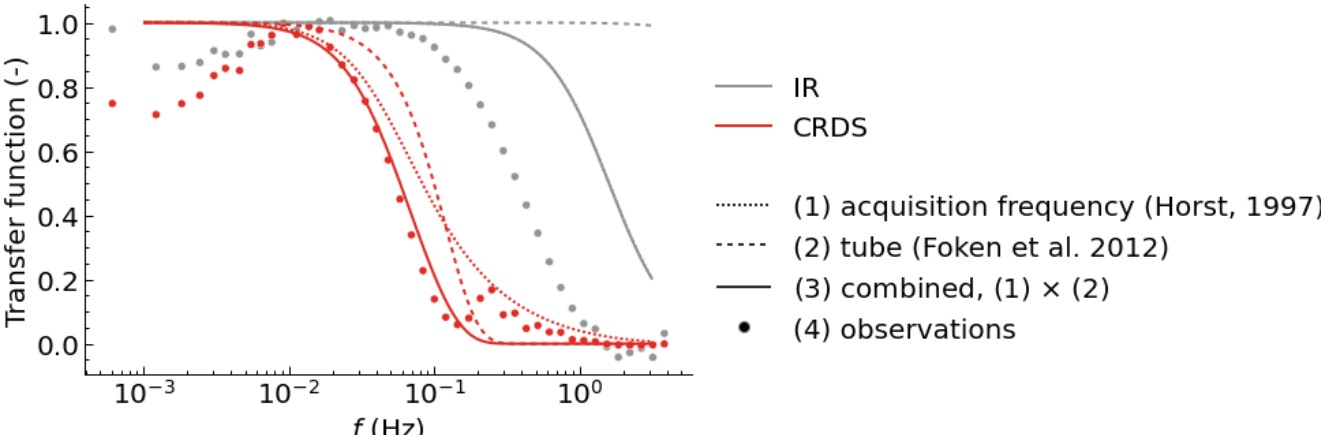

**Figure 16. Transfer functions computed for (1) the sensors acquisition frequencies (TF$_{acq}$, Horst, 1997), (2) the tube attenuation (TF$_{tube}$, Leuning and Moncrieff, 1990; Foken et al., 2012), (3) the combination of (1) and (2), and (4) the CO$_2$ observations, for the two setups at the tower. The Saclay atmospheric setup (CRDS) consists of 100 m sampling line with a 9.5 mm diameter, a sampling frequency of ~0.3 Hz and a flow rate of 12.7 lpm. Conventional ecosystem setup (IR) consists of 0.7 m sampling line with a 5.33 mm diameter, a sampling frequency of 10Hz and a flow rate of 15 lpm. Note that for the IR setup, the curves (1) and (3) are superposed.**

It's worth mentioning, in our study, we used a first-order filter fitted on in-situ data as a transfer function, following Fratini et al. (2012) and shared by other studies (Ibrom et al., 2007; Peltola et al., 2021). On the atmospheric tower configuration where the main attenuation arises from the tube length, the transfer function may take an exponential shape as proposed by Leuning and Moncrieff (1990) and Foken et al. (2012), and the fitting may not be perfect, as depicted in Figure 16. The effect was however evaluated to be negligible on the correction factor.

For a same transfer function, attenuation may change based on the cospectra dependence on measuring height, wind speed, and stability parameter (z/L). Specifically, increases in wind speed and stability parameter, or decreases in measuring height, are expected to shift a cospectrum towards higher frequencies, thereby enhancing attenuation for a given transfer function (Horst, 1997). Theoretical expectations of the attenuation factor from Horst et al. (1997) based on empirical cospectra agree very well with our measurements under unstable to near neutral conditions, but do not entirely align with our observations for

neutral and stable conditions (Figure 17). Indeed, surprisingly, we found that the attenuation remained stable or slightly decreased for z/L values over 0.2 in the case of IR and CRDS respectively. This contrasts with the prediction by Horst et al. (1997), which suggested an increase by a factor of 5 under very stable conditions. This difference needs further investigation.





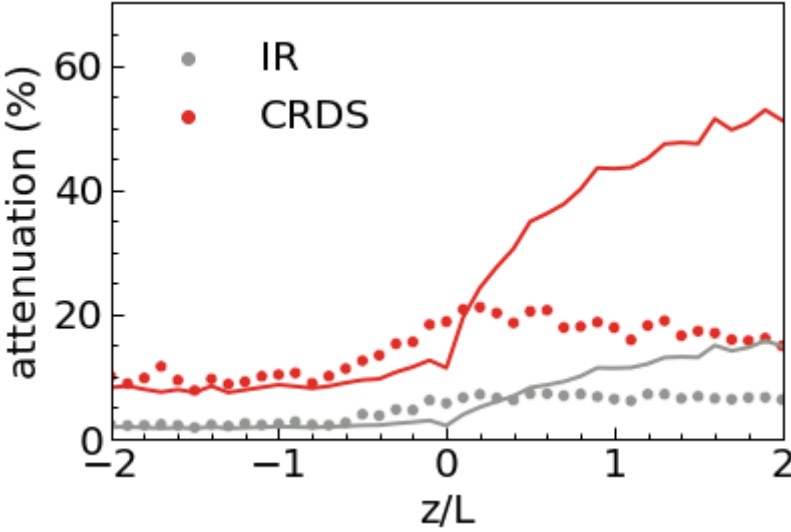

**Figure 17. Flux attenuation due to high frequency losses, theoretical as line and measured as points. The theoretical losses are computed from Horst (1997, eq. 11), using the measured first order time constant $\tau_c$ for the IR (0.5 s) and the CRDS (3s)**

More surprising is the HF losses found for $CO_2$ measured by the IR (LI-7200). On this supposedly conventional ecosystem setup (0.7 m heated tube, 5.33 inner diameter, 15 $Lmin^{-1}$ flow rate, 7 µm filter) attenuation and the transfer function were expected to be much smaller. The time constant of 0.5 s is equal to the cutting frequency (0.32 Hz) reported by Ibrom et al. (2007) for $CO_2$ with 50 m long tube, 8 mm diameter, and a flow rate of 20 L $min^{-1}$. We do not have strong evidence to explain this very large attenuation and we are bound to speculate that this may be due to the inlet filter. Indeed, we observed a very large HF loss for $H_2O$ with a $\tau_c$ ranging from 0.7 s for RH < 30% to 5 s at 70% RH and 50 s for RH larger than 80% (data not shown). This is the sign that the inlet did accumulate water vapour, most probably on the filter holding hygroscopic aerosols. Since $CO_2$ dissolves in water, the microscopic water accumulated in the tube may have buffered the $CO_2$ leading to a large attenuation.

The CRDS setup, however, exhibited relatively small attenuation. In comparison, Wintjen et al. (2020) reported a damping factor of around 16-22% for a 48 m tube with a 6 mm diameter, measuring reactive nitrogen at 10 Hz. Despite the longer tube length and slower analysers in our study compared to Wintjen et al., their slower flow rate (2.1 L min−1) and the expected stronger air-wall interactions for reactive nitrogen compounds may have contributed to the higher damping factor in their study. Correcting for high-frequency losses resulted in agreement between the IR and CRDS methods within 3%, maintaining the elevated $R^2$ of 0.94 (Figure 10). This demonstrates the high-frequency correction was able to correct for the losses.

**4.1.2. Storage and advection**

Storage fluxes amplitude were of the same order of magnitude as the turbulent fluxes, as also observed by Haszpra et al. (2005). This means that storage term is an essential component to include when looking into the diel surface flux pattern. In particular





we observe a clear negative storage flux in the morning during the summer months that may account for both the early onset of photosynthesis and the expansion of the mixed layer (Figure 7).

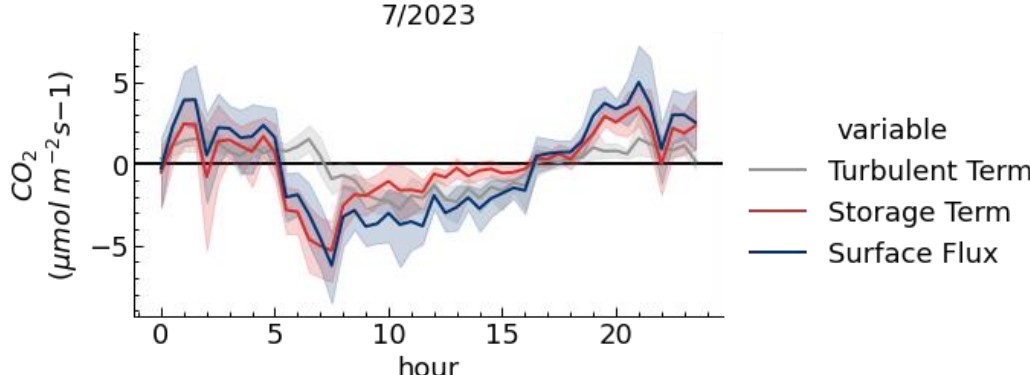

**Figure 18. Diel mean for July 2023 for $CO_2$ surface flux and turbulent and storage terms. Values for the CRDS (PICARRO G2401). Data points falling within the extreme 1% tail of the distribution were removed.**

Similarly, we observe a quite strong night-time respiration component after sunset that is not captured by the turbulent flux. Storage flux would therefore be essential to consider when partitioning $CO_2$ fluxes into biological uptake (photosynthesis) and emissions, which included both biological emission as respiration and anthropogenic emissions (Stoy et al. 2006; Tramontana et al. 2020; Coimbra et al. 2023).

In this work horizontal and vertical advection were assumed negligible, but in reality, they may not be. Indeed, daily mean
storage in the site gravitates around zero during July but shows a non-negligible variability for $CO_2$ in August and September and for all gases in January (Figure S6). In ideal conditions the storage flux should average 0 over 24 hours since what is stored at night should be destocked during the day. A non-zero daily storage may indicate horizontal advection.

Additionally, the chimney of the heating plant, ~600 m away from the tower, situated at a certain height, may bias the storage fluxes, which rely on gradient profiles along the tower. Indeed, if the plume from the heating plant emissions is measured
intermittently due to changes in wind directions, we may attribute or ignore storage fluxes where lateral advection is happening. Properly identifying such a process would require tracking the chimney plume with a 3D dispersion model and half-hourly resolution which was out of the scope of the present work.

Vertical advection may also exist, but is difficult to estimate and may even lead to erroneous corrections and implausible fluxes as observed by Haszpra et al. (2005). Vertical advection would even be neglected for very tall towers as proposed by Davis et
al. (2003), based on the assumption that synoptic scale processes should counterbalance vertical advection in the long-term.

**4.2. Plausibility of the measured fluxes**

Winds originating from the deciduous forest (West) exhibit expected seasonal variations, with $CO_2$ acting as a sink, and CO emissions being negligible in warmer months, transitioning to $CO_2$ and CO emissions in winter. Comparison with a mixed deciduous forest site at 50 km SE (FR-Fon ICOS site, e.g. Delpierre et al., 2016) shows a similar seasonality of the $CO_2$ fluxes,





a storage terms being slightly larger in FR-Sac than in FR-Fon, and a turbulent term notably smaller in FR-Sac (Figure S7). To explain the difference, we cannot exclude the possibility that the forests may behave differently simply because they are different ecosystems, however some aspects deserve to be mentioned. The surface flux in FR-Sac is expected to be smaller, since the forest only represents ~40% of the footprint. Moreover ~20% of the footprint is urban or traffic which may add a positive component to the flux decreasing its amplitude during the day (Figure 6). Advection could also play an additional role

in reducing surface fluxes inf FR-Sac, and storage may have been underestimated because of the reduced number of sampling heights. Furthermore, while this paper primarily examines the high-frequency aspect of the signal, it is important to note that the 30 min integration period did not allow to capture entirely the low frequencies of the fluxes, especially under unstable conditions as can be seen by the co-ogives slope being non-zero at the lowest frequencies (**Figure 9**). Recent study on urban tall towers also reported low frequency contribution for kinematic heat and $CO_2$ indicating the importance of low(frequency

corrections (Lan et al., 2024).

The presence of CO emissions during colder months may suggest contributions from activities beyond the forest, including traffic and nearby villages heating such as Villiers-le-Bâcle. We cannot exclude the possibility of indirect CO emissions from the forest through the oxidation of $CH_4$ and biogenic volatile organic compounds (BVOCs), though this should be limited since the trees lost their leaves. Additionally, soil microorganisms consume CO (Conrad, 1996), which makes CO a tracer to

distinguish soil respiration from anthropogenic emissions in the $CO_2$ emissions.



**Figure 19. Mean surface fluxes for $CO_2$ ($\mu mol m^{-2} s^{-1}$), $CH_4$ ($nmol m^{-2} s^{-1}$) and CO ($nmol m^{-2} s^{-1}$) by identified sectors. Monthly averages (a) and hourly mean for warmer JASO (b) and colder months DJ (c). Surface fluxes include turbulent and storage terms. Wind sectors are CEA (10°-100° without heating plant), Heating plant (20°-40°), Forest (247.5°-292.5°). Note in (a) November is interpolated.**

The emissions from the heating plant were clearly distinct, with significantly higher levels of $CO_2$ and $CH_4$ observed from ~30° wind direction compared to other directions (Figure 18). The high fluxes from point sources like the heating plant can induce non-stationarity. Using wavelets allows us to effectively measure these non-stationary fluxes. The $CO_2$, $CH_4$ and CO emissions from the heating plant wind sector increased to a maximum of 10000, 250 and 40 nmol m$^{-2}$ s$^{-1}$, respectively in January (Figure 18a), while they were not different from the other directions during the warmer months (Figure 18b). The $CO_2$ and $CH_4$ emissions further show a similar diel pattern during the colder month with two minima at 0h and 16h. The emission ratios computed from these data are $CO_2/CH_4$ ~ 40 and $CO_2/CO$ ~250, indicating a quite large loss of methane.





Interestingly, the diel pattern of CO during colder months followed expected traffic patterns, while this trend was observed for the CEA wind direction during warmer months.

### 4.3 Recommendations for atmospheric sites concerned by such a method

As of now, the ICOS network comprises 38 atmospheric sites, with 17 classified as class 1 and the remainder as class 2. The ecosystem centre is more extensive, encompassing 99 sites, including 18 class 1, 31 class 2, and the remaining associated sites. This count may further rise when incorporating other regional networks into consideration.

Not all atmospheric sites are adapted for flux measurements. EC towers prioritizes flat surfaces, slim towers and homogeneity whilst atmospheric towers may prioritize locations based on grid redundancy to improve atmospheric inversions. Slim towers with limited topography around are recommended for reliable measurements. Atmospheric measurements conducted in close proximity to large structures (e.g., just above domes) or in mountainous regions can introduce disturbances in the turbulence signal. This can lead to unreliable tilt angle correction and surface flux assessments. In some cases, flagging wind sectors not appropriated for EC measurements can be a straightforward solution. With that said, eddy covariance has successfully been used in mountainous landscape using appropriate tilt corrections (Matthews et al. 2017).

For atmospheric tower candidates interested in measuring flux, we recommend:

- Selecting at least one height for calculating fluxes through Eddy Covariance. For this decision a footprint estimation (Kljun et al., 2015) may be relevant.
- Including a high-frequency 3D anemometer on the chosen height(s).
- Evaluating the first-order filter time for tube and sensors, as showed here, to verify high-frequency attenuation is below an acceptable threshold, ~20%.
- Performing continuous mixing ratio measurements on the chosen height(s), either limiting profile measurements to to specific hours or with a separate set of instruments. This ensures low frequency signal for eddy covariance and profile can prioritise transition periods when fluxes may exhibit non-stationary behaviour or low turbulence.
- Evaluating the flow regime in the sampling tube, if possible increasing flow rate to guarantee turbulence ($Re > 2300$).
- Additional meteorological data (e.g.: precipitation, short and longwave incoming and outgoing radiation) and metadata (e.g.: forest type, crops, transport counting) pertinent for flux interpretation should be also collected.

Ensuring continuous measurements is crucial for wavelets, given their frequency resolution depends on the quantity of continuous data points. Certainly, profile measurements are often contemplated and doubling the instruments may not be feasible. Therefore, we recommended to restrict profile measurements to specific hours when the development of the boundary layer may overshadow the relevance of measuring flux close to the tower. These moments, typically during sunrise and sunset, provide valuable insights for both atmospheric and ecosystem (storage) perspectives. Furthermore, during these moments standard covariance would typically flag and disregard measurements, although to a lesser extent for wavelet-based eddy covariance, as depicted in Figure 8.



## 5. Conclusions


This study serves as a proof of concept for leveraging existing atmospheric towers to measure fluxes by simply adding a 3D anemometer. While eddy covariance on tall towers introduces challenges related to heterogeneity and storage effects, it mitigates concerns such as high-frequency attenuation. Comparing slow-response analysers with fast-response ones revealed very similar net fluxes across all stability conditions ($R^2>0.94$), indicating the viability of using slower instruments in this case.

It is important to note that our results focus solely on passive gases, as water was not considered due to air drying before measurement in the CRDS. For water, we could expect greater attenuation linked to tube length, as air-wall interactions of absorption and desorption are much stronger in water vapour (Massman and Ibrom, 2008). Similarly, we would not recommend measuring reactive gases, as their residence time might be too long (~60 s) for accurate eddy covariance measurements to be made. We also recommend using wavelet-based eddy covariance, as it enabled the exploitation of 18-34% more data compared

to conventional EC, allowing for the analysis of non-stationary fluxes. This was particularly evident in the case of a point source, such as a heating plant.

While many of the variables affecting attenuation are not under the researcher's control, limited choices remain for measurement height, tube dimensions, flow rate, and acquisition frequency. We recommend thus continuous gas measurements to be systematically done with high-frequency 3D anemometer, and a flow rate sufficiently large to ensure turbulent flows in

the sampling tube.

To calculate the surface flux, we estimated storage term. Although the storage term calculated using three heights provided useful estimations, caution is warranted due to potential biases from not measuring height at the same time, but also due to the limited number of heights sampled. Our results underscore the significance of the storage term, which was as large as the turbulent flux at the measurement height.

Analysing fluxes by wind direction revealed distinct patterns, particularly between the forest (W) and campus site (NE). Notably, emissions from a heating plant significantly influenced $CO_2$ and $CH_4$ fluxes in colder months, highlighting the importance of considering local sources. While our findings align with anticipated patterns across various land uses, accurately attributing fluxes to land uses would necessitate additional modelling efforts, which were beyond the scope of this study.

Overall, this study demonstrates the potential of expanding flux measurements through a relatively inexpensive

instrumentation addition, offering valuable insights for both ecosystem and atmospheric research. It further shows Eddy Covariance method has sufficiently matured so that we can use less-than-ideal instrumentation.

## Appendix A

Wavelet transform is a bandpass filter allowing decomposition of a time series into sub-series defined for a given frequency. The following steps briefly explain how to perform a frequency-resolved covariance using wavelets. More details can be found

in (Farge, 1992; Torrence and Compo, 1998; Farge and Schneider, 2001).




Any signal *f(t)* can be decomposed into different scales, which results in the signal itself once added up. The simplest example is the Reynolds decomposition that separates a time series into its mean and its instantaneous deviation:

$$f(t) = f'(t) + \overline{f(t)} \tag{A 1}$$

In Eq. 3, the mean, $\overline{f(t)}$, is the low-frequency component, with a frequency representative of 1/T, where T is the averaging period. Similarly, a time series can be decomposed into *J* sub-series, each representative of a band of frequencies *j*:

$$f(t) = \sum_{j=0}^{J} f(t,j) \tag{A 2}$$

The wavelet transform is a way to decompose the signal using a mother wavelet $\psi$, a finite wave function. Considering N discrete observations with a sampling period $\delta_t$, so that $t = n\,\delta_t$ where *n* is the time index, we can generate a family of wavelets normalized in L²-norm:

$$\psi_{n,j}(n') = s_j^{-1/2}\psi\left[\frac{(n'-n)\delta_t}{s_j}\right] \tag{A 3}$$

Where $s_j$ is the scaling factor, usually defined using a geometric progression with a maximum limited by the total sampling period $N\delta t : s_j = s_0 2^{j\delta_j}$, for $j = 0,1,...J$. Here, *J* is the size of the set of scales, $s_0$ is the smallest resolvable scale,
approximately $2\delta t$, and $\delta j$ is the scale factor. The convolution of the signal $f(n)$ with a scaled mother wavelet $\psi$, yields $W(n,j)$, the high pass wavelet coefficient for time series $f(n)$:

$$W(n,j) = \sum_{n'=0}^{N-1} f(n')\psi_{n,j}(n') \tag{A 4}$$

*W* is also named details to differentiate from approximation coefficient, result from a low pass filter. From *W* we can reconstruct the signal by first normalizing it:

$$\tilde{f}(n,j) = \frac{\delta j \delta t^{0.5}}{C_\delta \psi_0(0)}\Re\{W(n,j)\} \tag{A 5}$$

$C_\delta$ is a scale-independent reconstruction factor depending on the chosen mother wavelet function and $\tilde{f}(n,j)$ is the
decomposed signal defined in both time and frequency. The fully reconstructed signal is then found by summing to infinity the different frequencies:

$$f(n) = \sum_{j=0}^{\infty} \tilde{f}(n,j) \tag{A 6}$$





We can avoid summing to infinite by acknowledging the low pass filter, which yields an approximation coefficient which can be normalized to the low pass part of the signal named A:

$$f(n) = \sum_{j=0}^{J} \tilde{f}(n,j) + A_J \tag{A 7}$$

At the highest frequency $\sum_{j=0}^{0} \tilde{f}(n,j) = 0$ and $A_J = f(n)$. It becomes clear that (A 6) is a particular case for infinity. In the intermediate cases, $f(n)$ is the sum of all its components, $A$ is similar to a time average, and a finite sum of $\tilde{f}(n,j)$ is close to an instantaneous deviation.

$$f(n) - \frac{\sum_{n=0}^{T} f(n)}{T} \approx \sum_{f=\infty}^{1/T} \tilde{f}(n,f) \tag{A 8}$$

Where T in the averaging time (e.g. in seconds), $f$ is the frequency ($s_j^{-1}$, e.g. in Hz).

**On the (co)variance calculation using discrete and continuous wavelets**

We further explain the changes in the covariance calculation when using discrete or continuous wavelets. For variance consider the two variables are the same. For unidimensional variables, covariance calculation is straightforward, it is the mean product of two instantaneous deviations:

$$Cov_{x,y} = \overline{(x - \bar{x})(y - \bar{y})} \tag{A 9}$$

The frequency decomposition renders each variable bidimensional and as such their product should account for cross covariance.

$$Cov_{x,y} = \sum_{j=0}^{J} \overline{\tilde{x}_j \tilde{y}_j} + \sum_{k \neq j} \overline{\tilde{x}_j \tilde{y}_k} \tag{A 10}$$

The orthogonality of discrete wavelets implies independent frequencies, making $Cov_{x,y} = \sum_{j=0}^{J} \overline{\tilde{x}_j \tilde{y}_j}$. Thus, the cross-covariance problem only appears when using continuous wavelets. Some authors propose bypassing the reconstruction and calculating the covariance directly using the following equation (Torrence and Compo, 1998):

$$Cov_{x,y} = \frac{\delta j \delta t}{C_\delta N} \sum_{n=0}^{N-1} \sum_{j=0}^{J} W_x(n,j) W_y(n,j) \tag{A 11}$$

Note that the equation (A 11) cannot be retrieved from (A 5).

However, by postulating that the cross-covariance ($\sum_{k \neq j} \overline{\tilde{x}_j \tilde{y}_k}$) is proportional to the direct part ($\sum_{j=0}^{J} \overline{\tilde{x}_j \tilde{y}_j}$) at a factor determined by the mother wavelet, we can find an empirical factor, named $C_\varphi$, that corrects the covariance.





$$Cov_{x,y} = C_\varphi \sum_{j=0}^{J} \overline{\tilde{x}_J \tilde{y}_J} \tag{A 12}$$


Note that $C_\varphi$, like $C_\delta$, is only required for continuous wavelet decompositions in which the wavelet function is not an orthogonal base. The covariance correction factor $C_\varphi$ empirically found for Morlet and DOG wavelets (Coimbra et al., 2023) can be seen in Table A1.

**Table A1: Mother wavelets used in this study. Mother wavelet formula, $\psi(\eta)$, empirically derived factors, $C\delta$ and $\psi_0(0)$, from (Farge, 1992) and $C\varphi$ (Coimbra et al., 2023).**

| Name | Decomposition | $\psi(\eta)$ | $C_\delta$ | $\psi_0(0)$ | $C_\varphi$ |
|---|---|---|---|---|---|
| Morlet ($k_\psi$= 6) | Continuous | $e^{ik_\psi\eta}e^{-\frac{\eta^2}{2}}$ | 0.776 | $\pi^{-1/4}$ | 5.271 |
| DOG (m=2), also known as Marr or Mexican Hat | Continuous | $-1^m \dfrac{d^m}{d\eta^m}(e^{-\frac{\eta^2}{2}})$ | 3.541 | 0.867 | 16.568 |
| Daubechies ($k$=6) | Discrete | $(-1)^k a_{N-1-k}$ | 1 | 1 | 1 |





**Appendix B**

The theorical approach for high-frequency loss corrections requires defining a transfer function, TF, for each of the relevant
origins of frequency losses, $i$, and multiplying them to find the total transfer function, $TF_{total}$ (Moore, 1996):

$$TF_{total}(f) = \prod TF_i(f) \tag{B 1}$$

Note that TFs range between 0 and 1 and so the $TF_{ws}$ is driven by the most restrictive function for each frequency. Considering
only the attenuation from the air transport in the tube ($TF_{tube}$, Leuning and Moncrieff, 1990; Foken et al., 2012) and acquisition
rate ($TF_{acq}$, Horst, 1997):

$$TF_{tube} = \begin{cases} \exp\left\{-160 Re^{-\frac{1}{8}} \dfrac{\pi^2 r^5 f^2 L}{Q}\right\}, & Re < 2300 \\[3mm] \exp\left\{-\dfrac{\pi^3 r^4 f^2 L}{6 D_s Q}\right\}, & Re \geq 2300 \end{cases} \tag{B 2}$$

$$TF_{acq} = [1 + (2\pi f \tau_w)^2]^{-1/2} \times [1 + (2\pi f \tau_s)^2]^{-1/2} \tag{B 3}$$

Where, Re is Reynolds number (-), $r$ is tube radius (m), $f$ is the frequency (Hz), L is tube length (m), Q is volumetric flow rate
(m³s⁻¹), and $\tau$ is the first-order filter's time constant (s) where $\tau = (2\pi f_{acq})^{-1}$ and $f_{acq}$ is the acquisition frequency (Hz) for
vertical wind speed or scalar. Reynolds number is defined as $Re = \dfrac{2Q}{\pi r v}$, where $v$ is the kinematic viscosity.

Attenuation also depends on the cospectra. A theoretical approach is proposed in Horst (1997, eq. 11), where

$$\frac{\overline{w'c'}_{measured}}{\overline{w'c'}_{true}} = 1 + (2\pi n_{max} \tau_c \bar{u}/z)^{-\alpha} \tag{B 4}$$

Where $\bar{u}$ is the mean wind speed at height $z$, $n_{max}$ is 0.085 in case $z/L < 0$ else $2 - 1.915/(1 + 0.5z/L)$, and $\alpha$ is 7/8 for
$z/L < 0$ else 1.

**Code availability**

Code   used   in   the   analysis   presented   in   this   paper   is   available   online   and   can   be   accessed   at
https://github.com/pedrohenriquecoimbra

**Data availability**

ICOS data for FR-Sac and FR-Fon (forest site used for reference) can be downloaded from the carbon portal: data.icos-
cp.eu/portal.



**Author contribution**

PHHC, BL, and OL conceptualized the manuscript; PHHC, LB, and ML curated the data; PHHC and BL wrote the manuscript draft; OL, MR reviewed and edited the manuscript.

**Competing interests**

The authors declare that they have no conflict of interest.

**Disclaimer**

This project has received funding from the European Union's Horizon 2020 research and innovation programme under Grant Agreement No 101037319 (PAUL ICOS-Cities project).

**Acknowledgements**

The authors would like to thank ICOS, the ICOS Cities EU H2020 project (Grant No 101037319) for the support of this study. We thank Rainer Hilland for the support in hosting the Saclay data and Daniel Berveiller and Nicolas Delpierre for maintaining FR-Fon site. We also thank the SIRTA team, namely Simone Kotthaus for comments on the lidar data. We acknowledge using artificial intelligence for language checks.

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
