# Peer review of "Eddy-covariance with slow-response greenhouse gas analyser on tall towers: bridging atmospheric and ecosystem greenhouse gases networks"

_Atmospheric Measurement Techniques, 2024_

## Author Response (AR1)

**To Stefan Metzger Referee #1**

**On major comments**

Following recommendations on all 4 major comments we decided to focus the publication on the interoperability of slow and fast eddy covariance turbulence measurements including measurement height, transfer function, traditional quality flagging and wind directional interpretations. More intricate results on the storage flux, wavelets and footprint analyses are left to a separate publication.

**On specific comments**

Several specific comments became obsolete with the choice to drop the wavelet method and focusing on the turbulence flux in the results.

- **3.2**: Wavelets have been dropped from the manuscript. The reconciliation of resolving surface heterogeneity (footprint, Wavelet) while holding on to conventional mass conservation simplifications is an interesting question that deserves more attention than the current manuscript can focus on and will be the object of a separate publication.

- **3.2.1**: Following recommendation we moved the storage flux to a separate publication.

- **Lines 20, 90, 378, 608, 640**: We dropped the wavelets and so the statements regarding the requirement of stationarity for standard EC and wavelet-based EC will be further developed in a separate publication.

- **Line 170**: Mole-based dry mole fractions (mol mol^-1 dry air) were used.

- **Line 185**: Following recommendation we refocused the manuscript on the turbulent flux only and moving these explorations to a next publication.

- **Line 201**: Wilczak et al. (2001) was originally a citation error due to modifications in the original manuscript, thank you for noticing. The reviewed paper now uses planar fit and thus the Wilczak et al. (2001) was kept.

- **Line 230**: The wavelet was moved to a next publication. Further storage and advection terms were moved to discussion.

- **Line 289**: Added the "simple parameterization of".

- **Lines 318, 352**: The mentions to $H_2O$ were dropped. The slow instrument takes air previously dried, making $H_2O$ unusable and thus not an object of the present manuscript.

- **Line 345**: Figure 7 shows that during summer stable and unstable conditions were often the case while in winter it was mostly neutral. Added "summer nights" instead of "summer" and mentioned the figure to the reader.

- **Line 368**: Sentence changed to "We note that horizontal winds, over heterogeneous terrain in particular in stable and neutral conditions would favour horizontal advection".

- **Line 378**: The sentence was removed. To answer the question, ITC flagging test the development of turbulent conditions which are required to preserve the relationship between the variance of a turbulent quantity and its flux. The test, ITC, is a measure of the flux-variance similarity.

- **Line 434**: CO uncertainty added in the end of section 4.3.1 (revised manuscript).
- **Sections 4.1.2 and 4.2** Added mention that the negative flux due to expanding mixed layer is an artefact. Mention to partitioning in sections 4.1.2 (original manuscript) was moved to 5.2 (revised manuscript) as future perspectives.
- **Line 601**: We focus the revised manuscript on the turbulent flux, decreasing the references to surface flux and clarifying the terms missing to consider surface flux.
- **4.3, lines 646, 669**: Thank you for the positive feedback.
- **Line 640 following**: Paragraph revised removing wavelets.

**To Anonymous Referee #2**

- **Line 95:** Following recommendation we dropped the use of wavelets, to be used in a separate publication.
- **Lines 340 & 445:** Following recommendations from the reviewers, the manuscript was refocused on the interoperability of slow and fast eddy covariance turbulence measurements. Additional results on the footprint analyses are left to a separate publication. The uncertainties are thus mentioned but do not represent a relevant impact on any result or interpretation.
- **Lines 570:** Added mention to entrainment in section 2.2 (revised manuscript): "entrainment from the top of the atmospheric boundary layer may have only have significant impact on lower frequencies (Asanuma et al., 2007). Here, we assume entrainment plays a negligible role in the turbulent fluxes."

**To Anonymous Referee #3**

**Comments on technical aspects**

Spectral correction

- Power spectra added in supplementary material. The default option to remove 1Hz onwards was deactivated (in the reviewed manuscript). The revised manuscript uses the optimised response time, i.e. the correction from EddyPro as it is.
- P11L272: Corrected.
- Corrected eq. 12 where normalization is done with variance ($\sigma^2_s$), not standard deviation ($\sigma_s$).
- P27L545: At this height the very high frequency is in theory contributing less to the flux and thus the noise removal focus on this region is probably not attenuating the "true" signal. The power spectra (Figure S1) was added.
- A mention of the resampling strategy and its effects on FFT was added in section 5.1.1 (revised manuscript). We note that problems related to the resampling happen at frequencies higher than setup attenuation and have limited effect on the transfer functions and corrections.

Storage and advection

- Data was rerun to use planar fit and include vertical advection in the discussion see Figure 15.
- Added recommendation to alternate the levels every 5 minutes providing more samples within 30 min when doing profile measurements.

**Comments on Structure/Flow**

- The paragraph starting at P2L57 (original manuscript) was reworded to avoid parentheses and gain in fluidity.
- A new section 2 entitles "Requirements and constraints of slow and fast response analysers" freeing introduction from more technical details.
- Summarised paragraphs describing CO (P4L114, original manuscript) and CH4 (P4L123, original manuscript).

**On minor comments**

- P5L135: Clarified that the paper simply recommends adding a fast-response sonic anemometer by the side of an existing slow-response gas analyzer.
- Section 2.2.2 There was no filtering due to tower disturbance on the flow. It's a tall thin tower with measurements on the top. No major disturbances are expected, minor disturbances are possible from another anemometer and a Franklin antenna, both around 1 m away.
- Figure 4. Removed H2O from legend.
- Figure 5. Caption corrected to median. Added $25^{th}$ and $75^{th}$ percentile in figure.
- P15L339 – 340: Footprint percentages clarified.
- Added mention to uncertainty from footprint in methodology 3.4 (revised manuscript).
- P18L396: Using planar fit the ogive changed and so CH4 now reaches a plateau for near neutral and stable conditions.
- P18L397: For clarification we original line: "which after analysis was attributable to noise from this less sensitive instrument." into: "which after analysis was attributable to amplitude resolution from a signal with weak variance and a less sensitive instrument.".
- Figure 11. It referred to 95% confidence interval in the original manuscript. To facilitate understanding now it refers to standard deviation.
- P20L429: The air is dried before reaching the PICARRO. Water fluxes might then not be the focus on the interoperability.
- Figure 18. The new results using planar fit, instead of double rotation, increased turbulent fluxes during early morning. Results show a compensation between positive turbulent and negative storage terms, coming a near zero flux.
- P28L572: Modified accordingly.